

# Extending the Community Multiscale Air Quality (CMAQ) Modeling System to Hemispheric Scales: Overview of Process Considerations and Initial Applications

Rohit Mathur[1], Jia Xing[1,2], Robert Gilliam[1], Golam Sarwar[1], Christian Hogrefe[1], Jonathan Pleim[1], George Pouliot[1], Shawn Roselle[1], Tanya L. Spero[1], David C. Wong[1], Jeffrey Young[1]

[1]National Exposure Research Laboratory, Office of Research and Development, U.S. Environmental Protection Agency, Research Triangle Park, NC, USA
[2] School of Environment, Tsinghua University, Beijing, 100084, China

*Correspondence to*: Rohit Mathur (mathur.rohit@epa.gov)

**Abstract.** The Community Multiscale Air Quality (CMAQ) modeling system is extended to simulate ozone, particulate matter, and related precursor distributions throughout the Northern Hemisphere. Modelled processes were examined and enhanced to suitably represent the extended space and time scales for such applications. Hemispheric scale simulations with CMAQ and the Weather Research and Forecasting model are performed for multiple years. Model capabilities for a range of applications including episodic long-range pollutant transport, long-term trends in air pollution across the Northern Hemisphere, and air pollution-climate interactions are evaluated through detailed comparison with available surface, aloft, and remotely sensed observations. The expansion of CMAQ to simulate the hemispheric scales provides a framework to examine interactions between atmospheric processes occurring at various spatial and temporal scales with physical, chemical, and dynamical consistency.

**Keywords**: Hemispheric CMAQ, WRF, long-range transport, background pollution

## 1 Introduction

Comprehensive atmospheric chemistry-transport models must constantly evolve to address the increasing complexity arising from emerging applications that treat multi-pollutant interactions at urban to hemispheric spatial scales and hourly to annual temporal scales. To assist with the design of emission control strategies that yield compliance with more stringent air quality standards, such models must accurately simulate ambient pollutant levels across the entire spectrum ranging from background to extreme concentrations. The adverse impacts of airborne pollutants are not confined to a region or even a continent (NRC, 2009). Both observational (e.g., Andrea et al., 1988; Fishman et al., 1991; Jaffe et al., 1999; Zhang et al., 2008; Uno et al., 2009) and modeling studies (e.g., Jacob et al., 1999; Fiore et al., 2009; HTAP, 2010) have demonstrated that pollutants near the Earth's surface can be convectively lofted to higher altitudes where strong winds can efficiently transport them from one continent to another, thereby impacting air quality on intercontinental to global scales.



As air quality standards are tightened, the need to quantify the contributions of long-range transport on local pollution becomes increasingly important. Limited-area models such as the Community Multiscale Air Quality (CMAQ) (Byun and Schere, 2006; Foley et al., 2010; Appel et al., 2017) have played a central role in guiding the development and implementation of the National Ambient Air Quality Standards (NAAQS). These models are now being routinely applied to examine variability in surface-level air pollutants across the continental U.S. over annual cycles. Since transport is efficient in the free-troposphere and since simulations over continental scales and annual cycles provide sufficient opportunity for "atmospheric turn-over", i.e., exchange between the free-troposphere and the boundary-layer, it can be argued that accurate simulation of the variability in free-tropospheric pollutant concentrations is important for the model's ability to capture the variability in surface-level concentrations, especially at moderate-to-low concentrations. Based on typical advective time scales, it can further be postulated that in limited-area models, this free-tropospheric variability in simulated concentrations is largely dictated by the specification of lateral boundary conditions (LBC). This is exemplified in Figure 1 which illustrates the influence of LBC specification on simulated surface-level concentrations across a typical regional modeling domain covering the contiguous U.S.; the space and time varying LBCs themselves were derived from the Integrated Forecasting System of ECMWF (Flemming et al., 2015). In these calculations, three tracer species were added to CMAQ to track the ozone ($O_3$) LBC specified for three vertical zones: (i) surface to 750 mb, nominally representing the atmospheric boundary layer (BL), (ii) 750-250 mb representing the free-troposphere (FT), and (iii) 250 mb to 50 mb (the model top) representing possible stratospheric influences (cf. Mathur et al., 2008). These tracer species were subject to transport processes associated with 3-D advection, turbulent mixing in the vertical and horizontal diffusion, and cloud transport on resolved and sub-grid scales. The tracer species were deposited at the surface using space- and time-varying deposition velocity estimates for $O_3$, and they were also subjected to wet scavenging and rainout processes that mimic modelled $O_3$. If the tracer background is defined as the amount of the tracer imported into the regional domain, then the sum of the three simulated tracers can be viewed as the modelled background for this species. Since we did not include chemical sinks for the tracers and since the intent here is to assess the relative influence of LBC, we examine the distribution of the normalized concentrations (normalized by the domain maximum value across all seasons) in Figures 1a-d. Significant spatial and seasonal variability is seen in the estimated tracer background levels, with higher values in the high elevation regions of the inter-mountain west. Additionally, higher background levels are estimated in the warmer seasons, with highest levels during spring. More importantly, the free tropospheric concentrations dominate the surface background levels (Figures 1e-h). During summer, across most of the continental U.S. more than half (and up to 90%) of the tracer background originated in the FT. Though significant seasonal variability is noted in the FT fractional contributions to surface-level background concentrations, the contributions are still substantial during other seasons and expectedly the highest contributions are seen in the high elevation regions of the Western U.S. across all seasons. These results clearly illustrate the importance of accurately characterizing the long-range transport that occurs in the free troposphere and its influence on surface background pollution levels via subsidence and entrainment into the boundary layer. It can be expected that pollutants with atmospheric lifetimes greater than a few days would exhibit similar characteristics, thereby highlighting





the need to accurately characterize long-range transport influences on regional model simulations spanning seasonal to annual time-scales.

One approach to capture the effects of long-range transport in regional models is through deriving space- and time-varying LBC from global chemistry transport models. However, efforts linking regional and global scale models have met with mixed success because biases in the global model can propagate and influence regional calculations and often confound interpretation of model results (e.g., Tang et al., 2008; Schere et al., 2012). Additionally, inconsistencies in process representations, species mapping, and grid structures could also introduce errors in the model linkage if not examined and handled carefully. A modeling framework is thus needed wherein interactions between processes occurring at various spatial and temporal scales can be consistently examined. Expanding comprehensive regional models to the hemispheric scale enables a consistent representation of atmospheric processes across spatial and temporal scales. Motivated by this need, the applicability of the CMAQ modeling system has been extended to hemispheric scales through systematic investigation of key model processes and attributes influencing simulated distributions of $O_3$, fine particulate matter ($PM_{2.5}$), and precursor species. The hemispheric modeling system also facilitates examination of linked air pollution-climate across a region in context of the changing global atmosphere.

Section 2 overviews the key CMAQ model structural attributes and process representations that were refined to fully simulate the Northern Hemisphere. Section 3 summarizes a variety of applications with the hemispheric CMAQ configuration, highlights the model performance relative to a variety of observational data sets, and identifies aspects that would benefit from further model development. A variety of surface, aloft, and remotely sensed observations used to guide and evaluate the model changes are presented in sections 2 and 3. Section 4 summarizes the current model state and discusses future development and applications of the hemispheric CMAQ.

## 2 Model Setup and Process Enhancements

Atmospheric chemistry and transport of trace species occur across the continuum of spatial and temporal scales. For instance, transport across intercontinental to hemispheric scales occurs over timescales ranging from days to months, which influences the distribution of trace species with lifetimes within this range. Transport on these scales can also influence shorter lived radical budgets through chemical reactions involving intermediate-lived species, especially reservoir species such as organic nitrates. Thus, the expansion of CMAQ to hemispheric scales required re-examination of process representations and grid structures so that interactions amongst various processes occurring over the disparate scales is adequately captured. The key changes to CMAQ that were considered in this effort are summarized below.



## 2.1 Domain and Grid Configuration

CMAQ's governing three-dimensional equations for species mass conservation and moment dynamics (number, surface area, and volume) describing modes of particulate size distribution are cast in generalized coordinates (cf., Mathur et al., 2005; Byun and Schere, 2006). This formulation allows CMAQ to accommodate horizontal map projections and vertical coordinates from various meteorological models. This flexibility enables CMAQ to be used on a horizontal domain covering the Northern Hemisphere set on a polar stereographic projection (Figure 2a) without altering CMAQ or its input/output file structure. Current hemispheric applications have utilized a horizontal discretization of a 187x187 grid configuration with a grid spacing of 108km. Current regional modeling applications with CMAQ typically utilize 35 layers of variable thickness to resolve the model vertical extent between the surface and 50 mb. Longer-term calculations over the Northern Hemisphere must be able to capture potential impacts of stratosphere-troposphere exchange (STE) as well as that between the free troposphere and the BL. At altitudes above 10 km (Figure 2b), the 35-layer configuration has relatively coarse resolution with layer thickness >1.5 km and the top-most layer is nearly 4 km deep. To improve the representation of three-dimensional transport processes on modelled vertical profiles, the vertical resolution employed in hemispheric CMAQ calculations is increased. The revised layer structure uses 44 layers, with significantly finer resolution above the boundary layer (Figure 2b) to better represent long-range transport in the free troposphere, STE processes, and influences from cloud mixing both at the sub-grid- and resolved scale. The impacts of using these alternate layer configurations are illustrated in Figure 3 for a case where emissions only across the U.S. were zeroed out to isolate the impacts of model vertical resolution on representing the downward transport of pollutants in the region. Both the 35-layer and 44-layer model simulations were initialized with the same conditions in mid-February 2006, utilized a constant potential vorticity scaling to specify $O_3$ in the model top-layer (discussed further in Section 2.5), and were driven by meteorological information from the Weather Research and Forecasting (WRF) model simulations using the respective layer configurations (discussed in Section 2.2). Figure 3 shows systematically higher simulated $O_3$ below 10 km in the 35-layer configuration compared to the 44-layer configuration, indicating that the coarser vertical resolution will likely overestimate the downward transport of both long-range transport effects as well as stratospheric influences.

## 2.2 Coupling of WRF and CMAQ and Initialization

To minimize interpolation error and to avoid introducing mass imbalances, hemispheric simulations with CMAQ inherit the projection and grid structure from the WRF model, which provides the driving meteorological fields. In applications presented in Section 3, meteorological inputs for grid nudging used in WRF over the Northern Hemisphere domain were derived from the NCEP/NCAR Reanalysis data which has 2.5 degree spatial and 6-hour temporal resolution; other reanalysis products such as GFS can also be used instead. Surface reanalysis based on a fusion of the NCEP/NCAR Reanalysis and NCEP ADP Operational Global Surface Observations on the WRF grid using the NCAR distributed objective analysis tool Obsgrid (http://www2.mmm.ucar.edu/wrf/users/docs/user_guide_V3.1/ users_guide_chap7.htm#techniques) is used for the indirect soil moisture and temperature nudging in the Pleim-Xiu land surface model (Pleim and Gilliam, 2009). The WRF configuration





over the Northern Hemisphere also used MODIS land-use classification with 20 categories, RRTMg shortwave and longwave radiation scheme (Iacono et al., 2008), and the ACM2 PBL model (Pleim 2007). WRF's simulation of hourly surface temperature, relative humidity, wind speed and direction was evaluated by Xing et al. (2015a) through comparison with observations from NOAA's National Center for Environmental Information (NCEI) Integrated Surface Data and no significant

bias in the meteorological fields were detected. WRF and CMAQ can be run either in the traditional off-line sequential manner or in the coupled mode with or without aerosol feedback effects (Mathur et al., 2010; Wong et al., 2012).

Expectedly, the application over expanded space and time scales necessitates closer attention to model initialization, especially in the free troposphere wherein typical residence time for most atmospheric pollutants of concern are long enough so that

initial conditions can persist. If the FT is poorly represented, model predictions within the boundary layer will be adversely affected.  Thus, unlike regional simulations with CMAQ which are initialized with a prescribed vertical profiles for different species or with concentration fields derived from global chemistry-transport models, for hemispheric applications it is recommended that CMAQ be initialized to "clean" tropospheric background values and allowed to build up based on the model physics and chemistry. The impact of these different initializations are illustrated through comparisons of the cases denoted

Profile IC and Clean IC in Figure 4 which compares monthly mean model and observed $O_3$ profiles at Trinidad Head, CA. In the Profile IC case, a vertical $O_3$ profile that monotonically increased from 35 ppb at the surface to 100ppb at model top was used for initialization. The clean IC case initialized $O_3$ at 30 ppb through the model column.  In both cases, the hemispheric model simulations were initiated on 1 November 2015. Large overestimations in $O_3$ through the troposphere are noted for the Profile IC simulation and arise from the profile used to initialize $O_3$ in the mid-troposphere. In contrast in the Clean IC case

wherein the model was initialized to "clean" tropospheric background values and allowed to build up based on the model physics and chemistry, resulted in much better agreement with the measured profile during spring (Figure 4a); however, by summer, overestimations developed (discussed further in section 2.4.1). Note that the clean IC case also utilized a fixed potential vorticity scaling for $O_3$ at the model top, described further in Section 2.5. Also the similarity in simulated $O_3$ profiles for the Clean IC and Profile IC cases by August, suggests the diminishing impact of initialization.  Based on these results, the

inherent seasonality in atmospheric transport and chemistry, and practices employed in previous global chemistry-transport model applications (e.g., Fiore et al., 2009), a spin-up of 12 months from clean tropospheric conditions is recommended for new CMAQ applications over the Northern Hemisphere. Additional future studies would be helpful to further constrain this spin-up period recommendation.

## 2.3 Emissions

Specifying emissions across the Northern Hemisphere is challenging because the distributions and compositions of emissions across the globe are rapidly changing and because emissions are poorly quantified in many regions.  In addition, simulations of CMAQ at broader spatial scales are influenced by emissions from marine environments (which are less prominent in





regional/continental applications), and intercontinental transport of other sources (e.g., wind-blown dust). Changes to emissions used by CMAQ for the Northern Hemisphere application are described below.

### 2.3.1 Global Emission Inventories

Two primary sources of global emission estimates have been used in hemispheric CMAQ applications to date. The first is
based on global emission inventory compiled by Argonne National Laboratory in support of the ARCTAS pre-mission planning and includes estimates for anthropogenic, international shipping, and biomass burning (http://bio.cgrer.uiowa.edu/arctas/emission.html). This inventory was used in early testing of the hemispheric CMAQ model (e.g., Mathur et al., 2014) and will be referred to as the ARCTAS inventory in subsequent discussions. More recent applications have relied on year specific estimates from the EDGAR (Emission Database for Global Atmospheric Research, version 4.2;
European Commission, 2011) database which reports emissions for 17 anthropogenic sectors on a $0.1° \times 0.1°$ resolution grid. Since EDGARv4.2 provides only $PM_{10}$ emissions, $PM_{2.5}$ emissions were estimated by deriving the ratio of $PM_{2.5}$ to $PM_{10}$ from the 2000-2005 EDGAR HTAP (Hemispheric Transport of Air Pollution, version 1) inventory (Janssens-Maenhout et al, 2012) and then applying this ratio to partition EDGARv4.2 $PM_{10}$ emissions into $PM_{2.5}$ and $PM_{2.5-10}$ (Xing et al, 2015a). In applications to date, biogenic VOC (Guenther et al., 1995) and lightning NOx (Price et al, 1997) emissions were obtained from GEIA
(Global Emission Inventory Activity; http://www.geiacenter.org). The monthly biogenic VOC emissions were further temporalized to hourly resolution for each simulation day. Annual mean lightning $NO_x$ emissions were distributed evenly to each hour of each simulation day. Xing et al. (2015a) further describe the processing of EDGAR emissions for CMAQ, including temporalization of the annual estimates to hourly model inputs, vertical distributions of anthropogenic and lightning emissions, and speciation of $PM_{2.5}$ and NMVOC emissions to model primary aerosol constituents and gas-phase species. It
should also be noted that several efforts are underway to harmonize regional emission estimates and incorporate them into global emission inventories with improved spatial and temporal resolution (e.g., Janssens-Maenhout et al., 2015). Furthermore, the SMOKE modeling system typically used to prepare emissions for regional CMAQ applications has recently been updated to support hemispheric CMAQ applications to allow for a more streamlined implementation of the various emission processing steps described above (Eyth et al., 2016).

### 2.3.2 Wind-blown Dust

The wind-blown dust emission parameterization employed in CMAQ (Tong et al., 2008) was adapted for hemispheric applications by making two primary modifications. First, the mapping for land-use categories representing potentially erodible dust sources was updated to map to the categories of the MODIS land-use types used in the hemispheric WRF-CMAQ configuration. Second, the threshold friction velocity (above which dust emissions occur due to wind action) for desert regions
was reduced to mobilize sufficient episodic dust emissions over the Sahara. The original value for threshold friction velocity, derived from the work of Gillette et al. (1980), was based on data from the Mojave Desert. However, Li et al. (2007) suggest a much lower (about half) threshold friction velocity based on dust samples from the northern China desert. Fu et al. (2014)





found that the default threshold friction velocity for loose, fine grained soil with low surface roughness was too high for Asian dust sources and that reducing it to the Li et al. (2007) values yielded much better agreement of simulated airborne dust relative to observations. We found similar underestimations in PM$_{2.5}$ concentrations and AOD over the Sahara with the default values, and have thus followed an approach similar to Fu et al. (2014) in the hemispheric CMAQ applications presented here.

Concurrent with the development of this paper, a newer physics-based windblown dust emission parameterization was developed and implemented in CMAQ, and that parameterization includes a dynamic relation for the surface roughness length relevant to small-scale dust generation processes (Foroutan et al., 2017). The new dust emission parameterization is currently being tested for hemispheric applications and will be available in future public releases of CMAQ.

### 2.3.3 Emissions in Marine Environments

Natural emissions of particulate matter and gas-phase species from the oceans can impact air quality in coastal regions, influence global burdens of atmospheric trace species and radiative budgets, and modulate lifetimes of tropospheric O$_3$ thereby influencing its long-range transport. A detailed representation of sea-spray particle emissions and chemistry is already available in CMAQ (Kelly et al., 2010), and it can be used for hemispheric scale applications without any modifications.

Reactive halogen emissions can play an important role in dictating lifetimes of O$_3$ in marine environments. Parameterizations to estimate marine emissions of bromine and iodine containing compounds for the three categories (halocarbons, inorganic bromine, and inorganic iodine) were developed for inclusion in the hemispheric-CMAQ. The halocarbons include five bromo-carbons (CHBr$_3$, CH$_2$Br$_2$, CH$_2$BrCl, CHBrCl$_2$, CHBr$_2$Cl) and four iodocarbons (CH$_3$I, CH$_2$ICl, CH$_2$IBr, CH$_2$I$_2$). The halocarbon emissions are estimated using monthly average climatological chl-$a$ concentrations derived from the Moderate

Resolution Imaging Spectroradiometer (MODIS). Sarwar et al. (2015) provide details on estimating halogen emission and comparisons with other existing estimates.

### 2.4 Enhancements to Gas-Phase Chemistry

The 2005 Carbon Bond Mechanism with updated toluene chemistry (CB05TU; Sarwar et al., 2011), commonly used in regional CMAQ applications, was also used for initial hemispheric-scale applications. Important enhancements to CB05TU were

implemented to improve (1) its ability to represent multi-day chemistry associated with cycling of NO$_x$ through reservoir organic nitrate species in the mechanism, and (2) representing chemical sinks for tropospheric O$_3$ due to halogen mediated chemistry in marine environments. Additionally, the more detailed RACM2 mechanism has also been implemented (Sarwar et al., 2013) to facilitate its use in follow-on hemispheric applications.

### 2.4.1 Organic Nitrate Lifetime

Organic nitrates form during the atmospheric photo-degradation of hydrocarbons in the presence of nitrogen oxides (NO$_x$) through reactions of peroxy alkyl radicals (RO$_2$) with NO as well as through reactions with NO$_3$, and they act as a reservoir



for oxides of nitrogen. In the CB05TU mechanism, the species NTR is used to represent organic nitrates. Depending on its modelled lifetime, NTR can potentially redistribute $NO_x$ from source regions to $NO_x$-sensitive remote areas where additional ozone may be produced. Representing inert and reservoir organic nitrate species in condensed mechanisms used in chemistry-transport models is challenging (e.g., Kasibhatla et al., 1997) since they can dramatically influence simulated $O_3$ and $NO_y$

distributions. In the CB05TU implementation in CMAQ, the chemical sinks for NTR include photolysis (producing $NO_2$) and reaction with OH (producing $HNO_3$). Additionally, defining a Henry's law constant for a single lumped species representing several alkyl nitrates such as NTR is challenging. In previous CMAQ versions, the Henry's law constant for PAN was also used for NTR, resulting in its very slow removal either through scavenging by clouds or through dry deposition at the Earth's surface. However, the Henry's law constants for several alkyl nitrates and hydroxyalkyl nitrates have been suggested to be

much higher (some comparable to $HNO_3$), especially those that are of biogenic origin (Shepson et al., 1996; Treves and Rudich, 2003). On the hemispheric scale, organic nitrates formed from isoprene are the largest contributor to the simulated tropospheric NTR burden and can consequently modulate the simulated tropospheric $O_3$ burden. Based on recent work by Xie et al. (2013), we updated the rate constant for the NTR+OH reaction to that for isoprene nitrates. The Henry's law constant for NTR was also mapped to that of $HNO_3$, thereby enhancing wet scavenging of NTR. Additionally, the dry deposition velocity for NTR

was mapped to that for $HNO_3$. Collectively, these changes result not only in faster $NO_x$ recycling from NTR but also faster removal of NTR through the enhancement of its dry deposition and wet scavenging physical sinks.

The impacts of these changes to representing NTR in CMAQ on simulated $O_3$ distributions are illustrated in Figure 4, which presents a comparison of monthly mean profiles of simulated $O_3$ mixing ratios for various cases with ozonesonde

measurements at Trinidad Head, California, a site nominally representing inflow conditions to North America. The comparisons shown in Figure 4 illustrate the relatively large effects of modulating the resultant NTR burden on the simulated $O_3$ distribution through much of the lower to mid-troposphere, especially during summer when isoprene emissions are high. In limited-area calculations with the CB05TU mechanism, it is likely that the NTR produced is transported out of the regional domains before it can significantly alter $O_3$ production. However, over the spatial and temporal scales of northern hemispheric

calculations, $NO_x$ recycled from NTR can modulate the simulated background $O_3$; consequently, accurate characterization of its sources and sinks becomes critical. Thus the hemispheric calculations provide a framework for examining the role of various physical and chemical processes on atmospheric chemical budgets in a consistent manner.

Additional improvements to representing NTR in the CB05TU mechanism in CMAQ are underway. In particular, replacing

the single alkyl nitrate species (NTR) in CB05TU with seven species to better capture the range of chemical reactivity and Henry's law constants (and thus the physical sinks) is being investigated (Schwede et al., 2014; Appel et al., 2017). Early comparisons of this expanded treatment with the simpler changes discussed above suggest that the approximations invoked through mapping the OH reactivity to isoprene nitrates (from Xie et al., 2003) and mimicking NTR's wet and dry removal rates to $HNO_3$, yield similar simulated $O_3$ distributions to the ones obtained from the expanded treatment.



### 2.4.2 Representation of Marine Environments

More than half of the Northern Hemisphere is covered by oceans. To accurately represent the inter-continental transport of pollutants, it is important to accurately represent how continental air masses evolve as they traverse the vast oceanic regions. The fate of $O_3$ in marine environments directly affects inflow to continental regions and background $O_3$ concentrations. Though $O_3$ photolysis in the presence of high water vapor results in chemical $O_3$ loss and is well quantified, additional loss of $O_3$ in these environments through deposition as well as chemical reactions with halogens emitted from the ocean is expected (Vogt et al., 1999; Read et al., 2008), but still uncertain and not represented in most tropospheric models. In expanding CMAQ to hemispheric scales, particular attention was devoted to the role of deposition and halogen chemistry in marine environments, which can serve as sinks for $O_3$ exported from continental outflow and in transit to other regions via long-range transport. An enhanced $O_3$ deposition treatment that accounts for the interaction of iodide in seawater with $O_3$ was implemented (Sarwar et al., 2015) and found to increase deposition velocities in marine environments by an order of magnitude. In addition, the gas-phase chemical mechanisms were expanded to include 25 chemical reactions involving 7 chlorine species (Sarwar et al., 2012), 39 chemical reactions involving 14 bromine species, and 53 chemical reactions involving 17 iodine species (Sarwar et al., 2015).

### 2.4.3 Alternate Gas-Phase Mechanism

As discussed in Section 2.4.1, characterizing multi-day chemistry and long-lived reservoir species is important for representing the long-range transport of pollutants and their distributions on hemispheric scales. To enable practical model applications over extended spatial and temporal scales, the chemical mechanisms must be sufficiently condensed to run efficiently while faithfully representing the chemistry over the space and time scales modelled. However, the impacts on model predictions of using different condensation rules and assumptions on species lumping and intermediate compounds are largely unquantified. To enable such investigation in the future over the hemispheric scale, an alternate chemical mechanism, RACM2 has also been implemented and tested in the hemispheric CMAQ (Sarwar et al., 2013). The RACM2 mechanism is designed to simulate remote to polluted conditions from the Earth's surface through the upper troposphere (Goliff et al., 2013). It consists of 363 chemical reactions including 33 photolytic reactions among 120 chemical species.

### 2.5 Representation of Stratosphere-Troposphere Exchange

Though the role of cross-tropopause transport of $O_3$ is acknowledged as a significant contributor to the tropospheric $O_3$ budget, the distribution of $O_3$ in the troposphere that originates from the stratosphere is still uncertain. Tightening $O_3$ NAAQS and decreasing amounts of photo-chemically derived $O_3$ due to continuously declining anthropogenic precursor emissions across large parts of North America and Europe, now put greater emphasis on accurately characterizing the fraction of $O_3$ in the troposphere, especially at the surface, that is of stratospheric origin. For instance, Roelofs and Lelieveld (1997) using a global chemistry-transport model estimated that stratospheric contributions to surface $O_3$ varied between 10-60% depending on



season and location. Clearly this fraction varies spatially and seasonally in response to the tropopause height, and perhaps even more episodically, from deep intrusion events associated with weather patterns and frontal movement. Potential vorticity (PV) has been shown to be a robust indicator of air mass exchange between the stratosphere and the troposphere with strong positive correlation with $O_3$ and other trace species transported from the stratosphere to the upper troposphere (Danielsen, 1968).

Numerous modeling studies have used this correlation to develop scaling factors that specify $O_3$ in the modelled upper tropospheric/lower stratospheric (UTLS) based on estimated PV. The reported $O_3$/PV ratios (e.g., Ebel et al, 1991; Carmichael et al, 1998; McCaffery et al, 2004) however exhibit a wide range: 20-100 ppb/PVu (1 PV unit = $10^{-6}$ $m^2$ K $kg^{-1}$ $s^{-1}$), and vary as a function of location, altitude and season. Based on extensive ozonesonde measurements available during the summer of 2006 from the IONS network (http://croc.gsfc.nasa.gov/intexb/ions06.html) and PV fields from the WRF model matched to

the location and time of the ozone-sonde launch, we examined the UTLS $O_3$-PV correlation at sites across North America. At 12 sites with sufficient number (11 or greater) of launches during August 2006, strong linear correlations ($r^2 > 0.7$) were noted, with slopes of the linear regression varying between 20 and 39 ppb/PVu (Mathur et al., 2008). Based on this analysis, in the initial implementation of STE impacts on tropospheric $O_3$ in hemispheric CMAQ, we scale the space and time varying model estimated PV in top-most model layer with a scaling factor of 20 ppb/PVu to specify $O_3$ at the model top. This initial

conservative choice for the $O_3$/PV ratio was in part dictated by lack of additional information on seasonality and also by the relatively coarse model resolution in the UTLS. As indicated in Figure 3, layer configuration influences the representation of STE and subsequent simulation of 3-D $O_3$ distributions. Thus the initial conservative choice of 20ppb/PVu was motivated by the desire to reduce any possible effects of excessive and artificial downward entrainment associated with inadequate vertical model resolution.

To overcome some of these challenges and to develop a more robust representation of STE impacts, we have recently developed a dynamic $O_3$-PV function based on 21-year ozonesonde records from World Ozone and Ultraviolet Radiation Data Centre (WOUDC) with corresponding PV values from WRF-CMAQ simulation across the Northern Hemisphere from 1990 to 2010. Analyses of PV and ozonesonde data suggests strong spatial and seasonal variations of $O_3$/PV ratios which exhibits

large values in the upper layers and in high latitude regions, with highest values in spring and the lowest values in autumn over an annual cycle. The new generalized parameterization, detailed in Xing et al. (2016a), can dynamically represent $O_3$ in the UTLS across the Northern Hemisphere. The implementation of the new function significantly improves CMAQ's simulation of UTLS $O_3$ in both magnitude and seasonality compared to observations, which results in a more accurate simulation of the vertical distribution of $O_3$ across the Northern Hemisphere (Xing et al., 2016a). These can then be used to derive more realistic

vertically and temporally varying LBCs for regional nested model calculations.





## 3 Hemispheric-Scale Applications, Analysis, and Evaluation

The hemispheric CMAQ model is now being used for a variety of process-based air pollution studies across the Northern Hemisphere over seasonal (e.g., Mathur et al., 2014; Sarwar et al., 2014) to decadal (Xing et al., 2015a) time scales. In this section we present an overview of these diverse evolving applications with the hemispheric CMAQ model. In some instances, the applications have been detailed before, but the analysis summarized here builds upon that previous work and those distinctions are stated in the individual application discussion. Detailed comparisons of model predictions of pollutant concentrations (and radiative properties) with corresponding observations are conducted to establish credibility in the model's use in these applications that range from representing episodic long-range pollutant transport to quantifying long-term trends in air pollution across the Northern Hemisphere, to emerging applications examining air pollution-climate interactions. Model applications are performed over the hemispheric domain and 44-layer structure illustrated in Figure 2. The CMAQ configuration is based on version 5.0 (CMAQv5.0) with the process updates detailed in Section 2. Two sets of applications are analyzed and evaluated in the subsequent discussion: (1) a 21-year simulation over 1990-2010 (in section 3.5 and 3.6), and (2) process sensitivity studies for the calendar year 2006, which were each initialized in September 2005 using fields from the prior 21-year simulation set (in sections 3.1-3.4).

## 3.1 Comparing model predictions with measurements from the 2006 INTEX-B campaign

The Intercontinental Chemical Transport Experiment-B (INTEX-B) was a NASA-led, multi-partner atmospheric field experiment conducted in the spring of 2006. A major objective of the second phase of the campaign during 17 April–15 May 2006 was characterizing the long-range transport and evolution of Asian pollution and implications for air quality across western North America (Singh et al., 2009). Airborne measurements of a variety of trace species were made over the remote Pacific as well along the inflow region to western North America from extensively-instrumented aircrafts and provide a unique data set to test and evaluate the ability of hemispheric CMAQ to represent the 3-D structure of air pollutants as they are transported long distances across the Pacific to eventually impact U.S. background pollution levels.

The NASA DC8 flights during 17 April–1 May 2006 were based out of Honolulu, Hawaii, and sampled the sub-tropical Pacific, while the flights during 4–15 May 2006 based out of Anchorage, Alaska, sampled the troposphere over the sub-Arctic Pacific region. Figures 5a-d present comparisons of modelled and observed $O_3$ mixing ratios along selected flight paths of the DC8 aircraft; several of these flights were designed to sample aged and fresh Asian Pollution over the Pacific (Table 5a in Singh et al., 2009). Modelled mixing ratios were extracted by "flying" the aircraft through the 3-D modeling domain; the spatial locations of the aircraft were mapped to the model grid, whereas hourly model output was temporally interpolated to the time of the measurement. Figure 5 shows results from three different CMAQ configurations aimed at isolating the impacts of STE and marine halogen chemistry on simulated 3-D $O_3$ distributions. Differences between the simulations denoted "PV+Halogen" and "Halogen, NoPV" are used to estimate the $O_3$ sources due to modelled STE processes, while differences





between the simulations "PV+Halogen" and "PV, NoHalogen" help quantify the model $O_3$ sinks due to halogen chemistry in marine environments. Comparisons of model predictions from the "PV+Halogen" simulation with observations along the various flight paths suggest that the model exhibits skill in capturing the vertical variations in $O_3$ observed in the region. The simulation that did not employ any PV scaling (green trace in Figure 5) systematically underestimates $O_3$. The improved

comparisons with measurements along the different flight paths for the PV+Halogen simulation suggest that $O_3$ in the lower to mid troposphere in this region during this period is often influenced by sizeable contributions from the stratosphere, and these enhancements are generally captured by the simulation employing the PV scaling.

The model's ability to simulate the regionally averaged vertical profiles sampled by the aircraft over the subtropical and sub-

Arctic Pacific is illustrated in Figures 5e and 5f, respectively. In constructing these composite average vertical $O_3$ profiles, the observed and the modelled data were averaged within each 500 m vertical bin and over all the flights in that region; the figure also shows standard deviation for the observations. These vertical profiles represent the mean conditions that occurred over subtropical and sub-Arctic Pacific during the study period. The model tracks the composite average gradients within the lower and upper troposphere in these regions and accurately simulates that there is higher $O_3$ in the sub-Arctic Pacific upper

troposphere relative to the subtropical Pacific. Also apparent in these comparisons is the systematic and large underestimation of $O_3$ throughout the troposphere in the simulation that did not account for any contributions due to STE processes. The much closer agreement of the observed composite profile with that derived from the simulations with the PV scaling further suggest that on average ~10 ppb (or greater) of the $O_3$ in the troposphere over the Pacific during Spring could be of stratospheric origin. Thus $O_3$ in air masses entering western North America is comprised of both anthropogenic contributions due to long-range

transport of aged pollution from Asia and central America as well as a natural stratospheric component. The composite $O_3$ vertical profile during this period derived from the hemispheric CMAQ is within the range of those predicted by other global atmospheric chemistry models illustrated in Singh et al. (2009).

Anthropogenic emissions from Asia are often lifted into the free troposphere and transported across the Pacific to North

America in 5-10 days (e.g., Jaffe et al., 1999). Enhancements to free tropospheric $SO_4^{2-}$ measurements over northwest North America have been attributed to Asian sources (e.g., Andreae et al., 1988; Barrie et al., 1994). Increasing $SO_2$ emissions in Asia could potentially increase the amount of $SO_4^{2-}$ imported to North America and impact local efforts to reduce regional haze and improve visibility in national parks. Consequently, developing tools that accurately characterize the long-range transport from source regions and the amount of aerosols (both natural and anthropogenic) in air masses imported into a region

is needed. To assess the ability of the hemispheric CMAQ model to represent airborne $SO_4^{2-}$ levels and gradients off the Pacific coast of North America, we compared model predictions of $SO_4^{2-}$ (total of $SO_4^{2-}$ in the Aitken and accumulation modes) distributions with measurements taken during the INTEX-B study. In addition to $SO_4^{2-}$ measurements from bulk aerosol filters on the DC8, measurements from the particle into liquid sampler (PILS) on board the C-130 aircraft (11 flights) were also analyzed to evaluate the simulated $SO_4^{2-}$ distributions within both the boundary layer and free-troposphere over the eastern





Pacific and western North America. Analysis of the evolution of the MODIS aerosol optical depth (AOD) retrievals during mid-April 2006 (van Donkelaar et al., 2008) documents the development and transport of an Asian plume to western north America, and transects of C-130 flight on 21 April 2006 during 2000-2300 UTC sampled this plume in the free troposphere off the coast of western U.S. Figure 6 illustrates the simulation of this episodic Asian plume transport event. The flight path

(color coded by UTC time) and sampling region are shown in Figure 6a. Simulated transport features of the Asian $SO_4^{2-}$ plume in the model layer at approximately 4 km altitude at 2100 UTC on 21 April are illustrated in Figure 6b while Figure 6c presents space and time matched comparisons of the model results with measurements along the C-130 flight path. Both the MODIS retrievals (in van Donkelaar et al., 2008) and model simulations in Figure 6b show the export of $SO_4^{2-}$ from East Asia and its eastward transport across the Pacific Ocean to the western coast of North America. As illustrated in Figure 6c, $SO_4^{2-}$ levels >1

µg m$^{-3}$ were often measured in the free troposphere. The comparisons in Figure 6c further show that CMAQ captures the $SO_4^{2-}$ enhancements in the free troposphere associated with this episodic event.

Comparisons of campaign average composite vertical profiles for $SO_4^{2-}$ for all the DC8 and C-130 flights are shown in Figures 7a and 7b, respectively. Relative to the observations, CMAQ tends to overestimate mean $SO_4^{2-}$ levels especially in the lower

troposphere as seen in the comparisons with the bulk filter measurements from the DC8. It should be noted that the C-130 PILS measurements represent $SO_4^{2-}$ mass only for particles size <1µm, while the model values which are total mass in the Aitken and accumulation modes, nominally represents particle sizes <2.5 µm. This discrepancy in particle size cut-offs between the measured and modelled $SO_4^{2-}$ in part contributed to the systematic overestimation relative to the C-130 PILS measurements. In their comparisons with model results, van Donkelaar et al. (2008) for instance used a factor of 1.4 to scale the PILS $SO_4^{2-}$

during INTEX-B to account for particle size restrictions. Using a similar scaling here would result in a much closer comparison of the composite measured and modelled $SO_4^{2-}$ profiles in Figure 7.

### 3.2 Episodic inter-continental transport of Saharan dust and impact on U.S. air quality

Some of the earliest recognition of long-range transport of air pollutants, ranging back almost a century, was through observations of inter-continental transport of dust (Husar, 2004). North Africa is one of the largest sources of wind-blown dust,

and the frequent transport of Saharan dust across the North Atlantic Ocean to the Caribbean has long been studied (e.g., Prospero and Carlson, 1972). Trans-Atlantic transport of major Saharan dust outbreaks can episodically influence tropospheric particulate matter loading in the southeastern U.S. The ability of the hemispheric CMAQ to simulate such long-range transport events is investigated through analysis of a Saharan dust transport event in late July-early August, 2006. The simulated development and trans-Atlantic transport of a Saharan dust plume during this period is illustrated in Figure 8, which presents

daily average enhancements in PM$_{2.5}$ concentrations attributable to wind-blown dust. Large amounts of dust lofted into the atmosphere were transported west across the Atlantic, eventually impacting Gulf coast region of the U.S. Surface-level PM$_{2.5}$ measured in the U.S. Gulf states showed enhanced values as seen in the average concentrations across monitoring sites in Florida (Figure 9b).



A demonstration of CMAQ's ability to simulate episodic long-range Saharan dust transport is shown using comparisons with surface-level $PM_{2.5}$ measurements at the AQS monitors in the Gulf states (Figure 9). The average change in bias in modelled $PM_{2.5}$ between simulations with and without dust emissions is shown in Figure 9a, which indicates a reduction in bias in the

simulation incorporating the impact of Saharan dust emissions and transport. Collectively, the analysis here and in section 3.1 demonstrate that the hemispheric CMAQ modeling system can represent with reasonable skill the impacts of episodic trans-Atlantic (Figures 8 and 9) and trans-Pacific (Figures 6 and 7) transport events on air pollution over North America.

**3.3 Assessing the influence of stratosphere-troposphere exchange on surface $O_3$**

The analysis of 3D $O_3$ distributions from model sensitivity simulations relative to aircraft measurements, discussed in section

3.1, indicated the influence of STE processes on tropospheric $O_3$ distributions over the Pacific during the INTEX-B study period. To further analyze impacts of STE on tropospheric and surface-level $O_3$ over different seasons and regions, two simulations for the calendar year 2006 were conducted with the hemispheric CMAQ: with and without the dynamic PV-scaling approach discussed in section 2.5. Figure 10 presents the simulated seasonal average influence of STE processes on daily maximum 8-hour average (DM8) $O_3$, estimated as the difference between the simulations with and without the dynamic PV-

scaling parameterization. As can be seen, the amount of $O_3$ at the surface that is of stratospheric origin varies substantially both spatially as well as seasonally. As expected, high-latitude regions typically show greater influence of STE at the surface. Also the contributions to surface $O_3$ from STE are much higher during Spring and Winter when height of the tropopause is lower (e.g., Elbern et al., 1998) and the stratospheric influence can penetrate far down to the lower troposphere (e.g., Wang et al., 2002).

Figure 11 presents an evaluation of the PV-scaling parameterization for representing the seasonal impacts of STE processes on surface $DM8O_3$ relative to measurements from the CASTNET monitoring network in the U.S. A third simulation of 2006 was conducted using a *constant* $O_3$-PV scaling factor of 20 ppb/PV unit rather than the dynamic scaling approach. The model-estimated stratospheric contribution to surface $DM8O_3$ at the CASTNET locations can be estimated as the difference between

the $DM8O_3$ from the simulations with and without the dynamic PV scaling. The bias in the $DM8O_3$ predictions was computed at each location for the simulations with constant-PV and dynamic-PV parameterization. A reduction in bias between these two simulations is a relative measure of the improvements in surface $O_3$ predictions from using the dynamic-PV parameterization. Figure 11 correlates the seasonal average of this bias change with the estimated stratospheric contribution. The calculated seasonal means at each location, were restricted to days with observed $DM8O_3$ >40 ppb. This helps screen out

days where low $O_3$ may not be captured due to model grid resolution or other process limitations, and limited the analysis to periods where STE influences were likely greater. A strong linear relationship is noted in Figure 11 between the bias change and estimated stratospheric contribution. Across all seasons and at most locations, the dynamic-PV parameterization reduced the bias in predicted surface $DM8O_3$ relative to the constant-PV scaling. More importantly, when the estimated stratospheric



contribution to surface $O_3$ is high, greater reductions in model $DM8O_3$ error are realized through the use of the dynamic-PV scaling parameterization, demonstrating the ability of the PV-based parameterization in representing the effects of STE on surface $O_3$ levels and its seasonal and spatial variability. Additionally, the improvements in model predictions (i.e., reduction in model bias) of $DM8O_3$ are also greater during Spring and Winter when the stratospheric contributions are higher (Figure

10).  These evaluation results help build further confidence in the use of the dynamic-PV scaling parameterization in the hemispheric CMAQ model and for representing the impact of STE processes on surface $O_3$ levels.

**3.4 Comparison of $O_3$ predictions using the RACM and CB05 mechanisms**

As mentioned in section 2.4.3, the RACM2 is also available as an alternate and more detailed representation of gas-phase

atmospheric chemistry for hemispheric-scale CMAQ applications. A detailed comparison of the CB05TU and RACM2 predictions for *regional scale* applications over the continental U.S.  was described in Sarwar et al. (2013). A brief summary of comparisons of tropospheric $O_3$ predictions using the two mechanisms in hemispheric CMAQ is presented in Figure 12. The simulations were conducted for May-August, 2006 and were initialized using chemical fields from an existing longer-term simulation. Differences in predictions of surface-level monthly mean $O_3$ mixing ratios across the Northern Hemisphere

using the RACM2 and CB05TU mechanisms are illustrated in Figure 12a. In the simulation using the RACM2, higher $O_3$ is noted in polluted regions (regions with higher $NO_x$ in Figure 13e), but lower values are seen in the remote areas.  These differences arise primarily due to higher rates of $NO_x$ recycling from organic nitrates and more active organic chemistry in RACM2.

To further assess the impacts of using different chemical mechanisms on 3-D $O_3$ predictions, modelled $O_3$ distributions were compared with ozonesonde measurements at Sable Island, Nova Scotia; Trinidad Head, California; and Hilo, Hawaii. Comparisons of monthly mean $O_3$ vertical profiles simulated using different chemical mechanisms with corresponding observed profiles are shown in Figures 12b-d. Also shown are predictions with a model configuration in which the RACM2 mechanism was augmented with the halogen chemistry described in section 2.4.2. At Sable Island, which often receives

outflow from the U.S. northeast corridor, RACM2 over predicts $O_3$ at lower altitudes. The higher $O_3$ relative to CB05TU in the North American outflow is likely due to the faster $NO_x$ recycling in RACM2 as discussed earlier. At Trinidad Head, both RACM2 and CB05TU overestimate $O_3$ near the surface, though RACM2 is closer to the observations at altitudes of 1000-3000 m. In contrast at Hilo, CB05TU overestimates $O_3$ and RACM2 is much closer to the observed profile. In general, the addition of halogen chemistry in RACM2 helps reduce the overestimation at lower altitudes.  At altitudes > 1km, the RACM2

$O_3$ predictions are generally in closer agreement with the observations at all three sites. These mixed performance results indicate that neither mechanism is necessarily better suited over the other for hemispheric scale calculations. Nevertheless, analysis with both the CB05TU and RACM2 demonstrate the importance of $NO_x$-recycling from isoprene nitrates and halogen





chemistry on simulated $O_3$ distributions. Additional analyses of $NO_y$ partitioning and $HO_x$ predictions is needed to gain further insights on the reasons for the differences between the behaviors of the two mechanisms.

### 3.5 Simulating long-term trends in tropospheric composition

Over the past two decades significant and contrasting changes have occurred in anthropogenic air pollutant emissions across the globe. Emissions control measures implemented in North America and western Europe have led to improvements in air quality in these regions. In contrast, due to increasing energy demand associated with rapidly growing economies and population, many regions in Asia and Africa are experiencing a dramatic increase in emissions of pollutants. These spatially heterogeneous emission trends across the globe have not only resulted in contrasting changes in human exposure levels to air

pollution (e.g., Wang et al., 2017), but are likely impacting long-range transport patterns and influencing background air pollution levels in receptor regions. Accurate characterization of these changes in the chemical state of the troposphere (and potential influences on atmospheric dynamics) is needed to guide future control measures aimed at protecting human and environmental health. To assess these contrasting changes in air pollution levels, the hemispheric CMAQ was used to simulate trends in air quality across the Northern Hemisphere over a 21-year period (1990-2010). Year specific emission inputs were

derived from the Emission Database for Global Atmospheric Research (EDGAR, version 4.2) database (European Commission, 2011) as discussed earlier in Section 2.3.1 and detailed in Xing et al., (2015a).

Satellite-based tropospheric $NO_2$ measurements are now providing valuable observable information on the changing emission patterns and air quality across the globe (e.g., Richter et al., 2005; van der A et al., 2008). To determine if the model can

capture the impact of these changing emissions on tropospheric composition, trends in model tropospheric vertical column distributions (VCD) of $NO_2$ were compared with those derived from radiances measured by satellite instruments GOME (Global Ozone Monitoring Experiment) and SCIAMACHY (Scanning Imaging Absorption spectrometer for Atmospheric CartograpHY). GOME $NO_2$ observations are available during 1995-2003, while SCIAMACHY $NO_2$ retrievals are available since 2002. Figures 13a, 13b, 13d, and 13e compare annual mean tropospheric vertical column $NO_2$ for the calendar years

2003 and 2010 derived from SCIAMACHY retrievals and hemispheric CMAQ. Spatial distributions of the estimated 2003-2010 trends in $NO_2$ VCD from SCIAMACHY and the model are presented in Figures 13c and 13f, respectively.

Figure 13 shows that the spatial distributions of $NO_2$ VCD across the Northern Hemisphere are generally well correlated between CMAQ and SCIAMACHY, with higher $NO_2$ in the industrial and urban areas of North America, Europe, and Asia.

Some discrepancies are noted in central Africa where CMAQ simulates higher tropospheric $NO_2$ in the Central African Republic and its northern border with Chad. Trends in $NO_x$ emissions derived from the EDGAR inventory show a similar increasing trend in this region (see Figure 2b in Xing et al., 2015a) and indicates that this discrepancy is associated with the underlying emission data set used in these simulations. In contrast, SCIAMACHY distributions in the region show a signal





associated with biomass burning in the Savanna region both in 2003 and 2010 – the spatial extent of which is not captured by the model. Comparison of 2003-2010 trends in tropospheric $NO_2$ between the CMAQ simulations and SCHIAMACHY also indicate that the model captures increases in East China, and many cities in India and Middle East as well as the decreases across the eastern U.S., southern California, and western Europe. Figure 14 presents comparisons of time-series of regional-average monthly mean variations in tropospheric $NO_2$ simulated by the model with corresponding values based on the GOME and SCIAMACHY retrievals for three regions: East China, United States, and Europe (see Figure 3 of Xing et al., 2015a for sub-domain extents). The domain-mean seasonal variability in tropospheric $NO_2$ (as represented by the GOME and SCIAMACHY retrievals) is captured reasonably well by the model with a cool season maximum and warm season minimum. The model accurately simulates the amplitude of this variability for the U.S. as well as its inter-annual variability. For East Asia the model underestimates the peak values. In contrast for Europe, relative to both GOME and SCIAMACHY, the model consistently overestimates the summertime minimum values. Note that these simulations did not account for aerosol radiative feedback effects which, due to scattering and absorption of incoming solar radiation, reduce the amount of radiation impinging the Earth's surface. The resulting stabilization can reduce boundary layer ventilation and increase surface-level concentrations. As shown in Xing et al. (2015b, 2015c) these effects are especially important in polluted environment such as East Asia. Consequently, some of the underestimation in tropospheric $NO_2$ over East Asia during the cooler months (when particulate matter pollution is the highest) could also arise from ignoring the aerosol direct radiative effects in simulated concentrations.

In addition to $NO_x$, anthropogenic emissions of $SO_2$ and volatile organic compound (VOCs) have also been reduced significantly in the U.S. To assess the impact of these precursor emission changes on trends in concentrations of secondarily produced species, we compared model simulated trends in ambient $O_3$ and aerosol $SO_4^{2-}$ with those inferred from measurements at the CASTNET monitors. Figure 15 presents comparisons of model and observed trends in annual average daily maximum 8-hour average $O_3$ and annual average weekly-average $SO_4^{2-}$ at each CASTNET monitor site. Also shown in Figure 15 are results from an additional simulation with CMAQ that used a 36-km regional domain focused on the contiguous U.S. (see Figure 2). This regional simulation used an updated long-term emission inventory for the U.S. (Xing et al., 2013) and was driven by space and time varying lateral boundary conditions from the hemispheric CMAQ simulations for 1990-2010 (Gan et al., 2015a). Figure 15 shows that both the hemispheric-scale and the nested regional model capture the decreasing trends in both $DM8O_3$ and $SO_4^{2-}$ as well as the spatial variability in the magnitude of the trends across the CASTNET sites, though the hemispheric model tends to underestimate the magnitude of the trends. However, the finer resolution of the nested simulation in conjunction with the updated emission inventory better captures the observed trends in surface level $DM8O_3$ and $SO_4^{2-}$ as indicated by slopes closer to unity. This suggests the need to further explore finer resolution model calculations with the hemispheric CMAQ. As computing resources increase in the future it may be possible to conduct hemispheric scale simulations utilizing grid spacing finer than the 108 km utilized here.





## 3.6 Assessment of representation of aerosol direct radiative effects

Both aerosol scattering and absorption reduce the shortwave radiation impinging on the Earth's surface. The variability in surface solar radiation plays a prominent role in the Earth's climate system as it contributes to the modulation of the surface temperature, intensity of the hydrological cycle, and potentially the net ecosystem productivity (Wild, 2009). Observed trends in solar radiation reaching the Earth's surface are very likely associated with changes in aerosol and aerosol-precursor emissions governed by economic development and air pollution regulations, which have modulated the trends in regional and global tropospheric aerosol burden over the past two decades (cf. Wild, 2009; Streets et al., 2009). Surface solar radiation "dimming" and "brightening" effects respectively dampen and enhance the warming trends induced by greenhouse gasses, so it is essential to accurately characterize these trends and quantify the role of regional variability in tropospheric aerosol burdens on these trends.

Simulations over the Northern Hemisphere can also be conducted using a *two-way coupled WRF-CMAQ* configuration (Wong et al., 2012), where CMAQ-simulated aerosol composition and size distribution are used to estimate their optical properties which are then *fed back* to the WRF radiation module to influence the simulated radiation by WRF. Thus the effects of aerosol scattering and absorption of incoming radiation can further impact the simulated atmospheric dynamics (boundary layer heights, temperature, simulated resolved and sub-grid scale clouds), which then impact emission rates, transport and dispersion, deposition, and temperature and actinic flux dependent chemical rate constants. The aerosol optical properties in the two-way coupled WRF-CMAQ are calculated using the BHCOAT coated-sphere module approach (Bohren and Huffman, 1983), i.e., particles in the Aitken and accumulation model are assumed to have a core composed of elemental carbon with a shell coating of other species. The aerosol optics calculations in the WRF-CMAQ have been evaluated against field measurements as detailed in Gan et al. (2015b).

In addition to the long-term (1990-2010) simulations discussed in Section 3.5, additional feedback simulations were conducted over the Northern Hemisphere with the two-way coupled WRF-CMAQ configuration for the summer months (June, July and August) of this 21-year period. Comparison of these two sets of simulations (with and without aerosol feedbacks) provides an indication of the impact of the aerosol direct radiative effects (ADRE) and an assessment of its trends associated with the changing tropospheric aerosol burden over the multi-decadal period. Figure 16 examines the modelled and observed relationships between changes in aerosol optical depth (AOD; representing changes in the tropospheric aerosol burden) and changes in clear-sky surface shortwave radiation (SWR) using regional averages for East China, Europe, and East U.S. Figure 16 examines 2000-2010, when satellite-based data were available. Monthly regional averages of SWR and AOD were calculated for each of the summer months (June, July, and August). To minimize the influence of month-to-month variability, monthly averaged SWR and AOD were deseasonalized by subtracting the average of 11-year data for the corresponding month. Additionally, we used 24-hour-averaged SWR but AOD at noon (local-time) to be consistent with the observation-derived data



from satellite-retrievals (Xing et al., 2015b). The change in the SWR and AOD for each summer month in the 2001-2010 period was estimated relative to the corresponding year-2000 value, and the relationship between these changes is examined in Figure 16 for both model simulations with and without direct aerosol feedback effects. Also shown in Figure 16 is the corresponding observed relationship between changes in AOD and SWR derived from retrievals from the MODIS and the

Clouds and the Earth's Radiant Energy System (CERES; Wielicki et al., 1998) instruments, respectively. Note that the CERES mission estimates clear-sky surface SWR through radiative transfer calculations using satellite-retrieved surface, cloud, and aerosol properties as input (Kato et al., 2013), which have also been shown to agree with surface observations (Wild et al., 2013).

All three regions show a strong relationship between observed changes in AOD and clear-sky surface SWR, with reductions in SWR associated with increases in AOD and increases in SWR with reductions in AOD. This observational comparison clearly suggests that as tropospheric aerosol burden increases, scattering and absorption reduces the amount of surface SWR. The magnitude of these changes is comparatively larger for East China than for Europe and East U.S., due to the higher levels of tropospheric particulate matter in East China. In all three regions, the model simulation without direct aerosol feedback fails

to capture the changes in SWR and its association with AOD. In contrast the model simulation incorporating the aerosol direct radiative effects replicates the relationship between the observed AOD and SWR changes during the 2000-2010 period in all three regions as reflected by the higher $R^2$ and slopes of the linear regression closer to those inferred from the observed data. These results suggest that the hemispheric two-way coupled WRF-CMAQ configuration can represent the differing regional changes in surface SWR with contrasting changes in regional aerosol burden. Accordingly, this tool could also be used to

examine chemistry climate interactions on hemispheric to regional scales.

**4 Summary and Concluding Remarks**

The applicability of the Community Multiscale Air Quality (CMAQ) modeling system was extended to the entire Northern Hemisphere to enable consistent examination of interactions between atmospheric processes occurring at various spatial and temporal scales. Improvements were made in model process representation (stratospheric $O_3$ influences, representation of $NO_x$

recyling through organic nitrates, halogen chemistry in marine environments, deposition over water), structure (model vertical extent and layer resolution), and input data sets (allocation of global emission estimates). These improvements to CMAQ were investigated and evaluated through comparison of model predictions with measurements from surface, aloft, remote sensing and specialized field campaign platforms. Comparisons with measurements from the INTEX-B field campaign indicate that hemispheric CMAQ captures the mean variability in $O_3$ and $SO_4^{2-}$ distributions observed over the tropical and sub-Arctic

Pacific regions, and episodic transport of Asian pollution across the Pacific as indicated by comparisons of model and observed $SO_4^{2-}$ along specific flight tracks. The ability to capture the development and evolution of inter-continental transport (i.e., the lofting of pollutants in the source region, multi-day transport in the free troposphere and subsequent subsidence and mixing



down to the surface in receptor regions) is also demonstrated by evaluating a trans-Atlantic Saharan dust transport event and its contributions to elevated surface $PM_{2.5}$ in the U.S. Gulf region. These results suggest that regional CMAQ applications can now be driven by space and time varying lateral boundary conditions derived from consistent hemispheric applications, enabling examination of air quality across the U.S. in context of the changing global atmosphere.

Hemispheric CMAQ model can reproduce historical trends in tropospheric air pollution, as shown by comparing simulated results with surface and remote-sensing observation-derived records during 1990-2010. Trends in modelled tropospheric $NO_2$ vertical column distributions agree with those inferred from GOME and SCIAMACHY retrievals and indicate the contrasting and heterogeneous changes in emissions across the Northern Hemisphere, with increases in rapidly developing regions of Asia

and decreases in Europe and North America resulting from implementation of control measures. Additionally, comparisons with observed trends at the CASTNET monitors, indicate that though the model captures the resultant decreasing trends in surface-level air pollution (for $O_3$ and $SO_4^{2-}$) in the U.S., the current configuration underestimated (by 25-47%) the magnitude of the trend at some monitoring locations. The underestimation in the magnitude of the trend is however significantly reduced in a nested regional simulation utilizing finer horizontal grid resolution and updated historical regional emissions. The changing

emission patterns across the Northern Hemisphere will likely influence future long-range pollutant transport patterns and potentially impact background pollution levels in receptor regions. The hemispheric CMAQ provides a framework to explore such changing impacts on air pollution exposure. For instance, Wang et al. (2017) estimated trends in $PM_{2.5}$ premature mortality during 1990-2010 using hemispheric CMAQ predictions, and show that correlations between population and $PM_{2.5}$ have become weaker in Europe and North America due to air pollution controls but stronger in East Asia due to deteriorating

air quality.

Analysis of aerosol optical and radiative effects inferred from the two-way coupled WRF-CMAQ applications over the Northern Hemisphere also indicate the association between changing tropospheric aerosol burden and clear-sky surface shortwave radiation. In rapidly developing regions such as East Asia, the increasing tropospheric aerosol burden results in

greater scattering and absorption by aerosols, and that acts to reduce the amount of clear-sky surface shortwave radiation. In contrast, increases in clear-sky surface shortwave radiation are noted in regions with declining tropospheric aerosol burden, where emissions controls have been more active during that period. The two-way coupled WRF-CMAQ configuration that incorporates direct aerosol radiative effects captures these contrasting observed changes in clear sky shortwave radiation across the Northern Hemisphere during 2000-2010, with "brightening" in U.S. and western Europe and "dimming" in East China.

Using these modeling results, Xing et al. (2016b) show that because of reduced atmospheric mixing resulting from direct aerosol feedbacks, air pollutants become more concentrated locally, especially in highly polluted and populated regions. Thus modulation of air pollution due to aerosol direct effects also translates into an additional human health dividend in regions (e.g., U.S. and Europe) with air pollution control measures but a penalty for regions (e.g., East Asia) witnessing rapid deterioration in air quality.



Analysis of three-dimensional $O_3$ distributions across the Northern Hemisphere from model sensitivity simulations and comparisons with surface and aloft measurements also highlight the need to better quantify the contribution of STE processes on surface $O_3$. A non-trivial contribution of up to ~10 ppb from the stratosphere to seasonal mean surface-level $O_3$ mixing

ratios is inferred from the current applications. An accurate characterization of this contribution is essential for source attribution of background $O_3$. Since measurements of 3-D $O_3$ distributions alone are insufficient to directly quantify this contribution, model estimates need to be better constrained. To that end additional CMAQ simulations that explore the sensitivity of STE process representation to model vertical extent and vertical grid resolution are warranted. Model calculations presented here also indicate the possible influence of horizontal grid resolution on model evaluation results.

Hemispheric CMAQ simulations to date have employed a horizontal grid resolution of 108 km, which is insufficient to resolve local gradients. Emerging environmental problems will likely require the simultaneous characterization of air pollution from local-to-global scales. Variable resolution nonuniform grids can simultaneously and accurately resolve local gradients and large-scale features in air pollutant distributions (e.g., Odman and Russell, 1991; Mathur et al., 1992; Srivastava et al., 2000). The emergence of variable resolution atmospheric dynamical models (e.g., Skamarock et al., 2012) provides opportunities to

develop comprehensive atmospheric modeling systems that seamlessly represent the scale interactions from urban to global scales. The use of such approaches could improve the representation of scale interactions in air pollution modeling.

Several efforts are underway to harmonize regional emission estimates and incorporate them into global emission inventories with improved spatial and temporal resolution (e.g., Janssens-Maenhout et al., 2015). It can be expected that future

improvements in performance of hemispheric CMAQ will also be realized through improvements in these underlying global emission inventories used to drive model calculations. Additional improvements in sector specific emissions and additional details on chemical speciation of the emissions will also lend themselves to the use of more detailed chemical mechanisms such as RACM2 that explicitly treat the chemistry of longer-lived species (e.g., acetone) that are important for chemistry of the upper troposphere, and help further assess the relative benefits of the use of different chemical mechanisms at hemispheric

scales. Predictions of a variety of atmospheric pollutant species from hemispheric CMAQ are also being compared to those from other modeling systems (Hogrefe et al., 2015) through activities of Air Quality Model Evaluation International Initiative (AQMEII). The adequacies and inadequacies of the lateral boundary conditions derived from hemispheric CMAQ to drive regional CMAQ simulations are further being analyzed through comparisons with those from other large scale models and observations (Hogrefe et al., 2017) and will also guide the future evolution of the hemispheric CMAQ.

**Acknowledgements**: We gratefully acknowledge the free availability and use of observational data sets from the INTEX-B field study, CASTNET, AQS, IONS, and WOUDC networks; remote sensing retrievals from GOME, SCIAMACHY, MODIS,



and CERES; and global emission inventories from EDGAR, GEIA, and ARCTAS efforts. We thank Havala Pye and Barron Henderson for comments and suggestions on the initial version of this manuscript.

**Disclaimer**: The views expressed in this paper are those of the authors and do not necessarily represent the view or policies of the U.S. Environmental Protection Agency.

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





**Figure 1**: Impact of lateral boundary conditions (LBC) on simulated seasonal surface-level concentrations. (a-d): Spatial variation in seasonal-mean surface concentrations normalized by the maximum value within the model domain across all seasons. (e-h): Fractional contribution of free-tropospheric (FT) LBCs (specified between 750-250mb) to the total LBC-derived concentrations at the surface. Seasons are defined as: Winter (December-February), Spring (March-May), Summer (June-August), Fall (September-November).





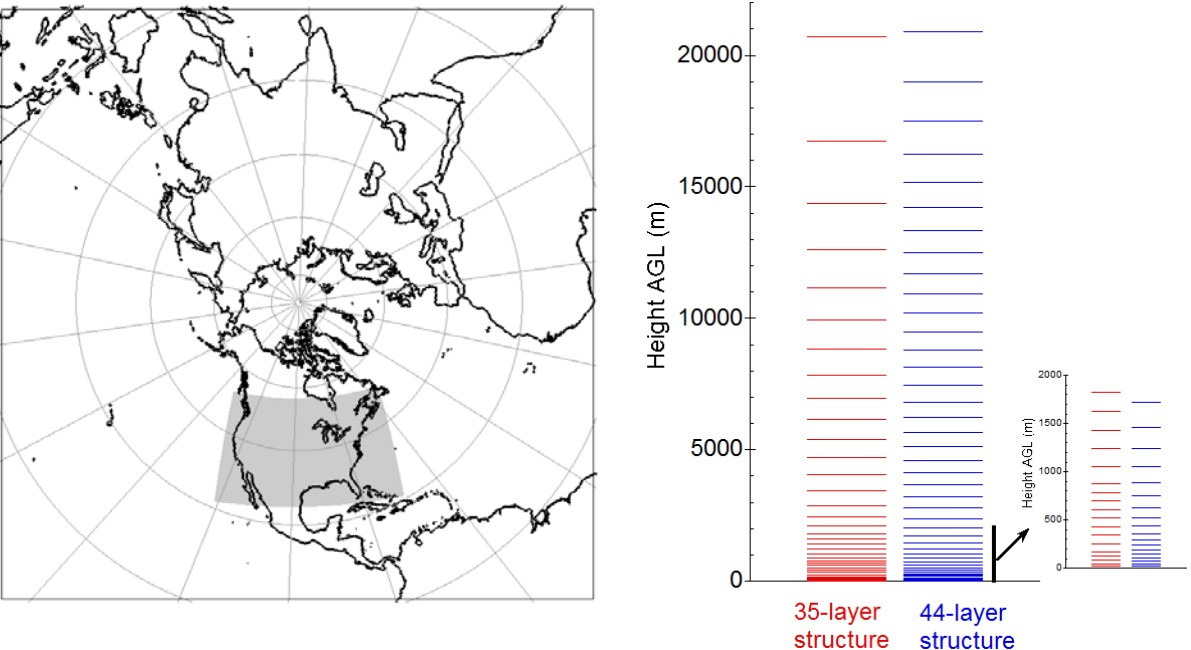

**Figure 2**: Left: The Northern Hemisphere modeling domain discretized using a 108 km resolution grid. The shaded region shows the extent of the typical nested Continental U.S. nested domain discretized using a 12 km resolution horizontal grid.

5    Right: Comparison of two layer configurations used to discretize the vertical extent ranging from the surface to 50 mb.





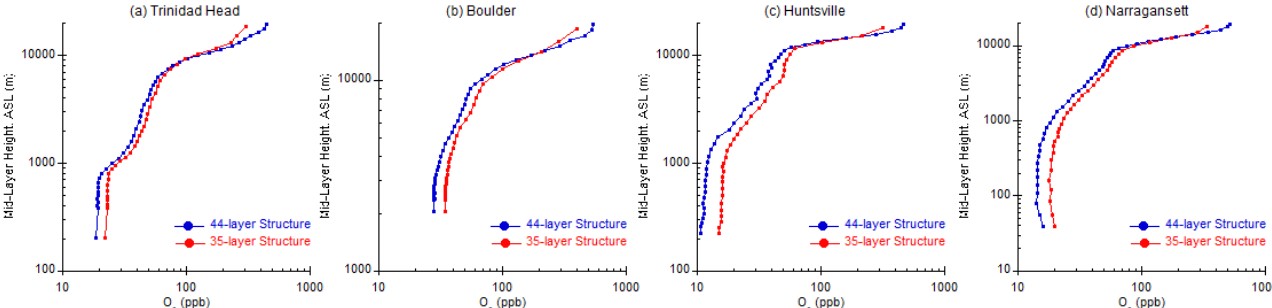

**Figure 3**: Impact of layer configuration on simulated mean $O_3$ vertical profiles for August 2006 at selected locations for a case involving zero-out of emissions across the U.S.



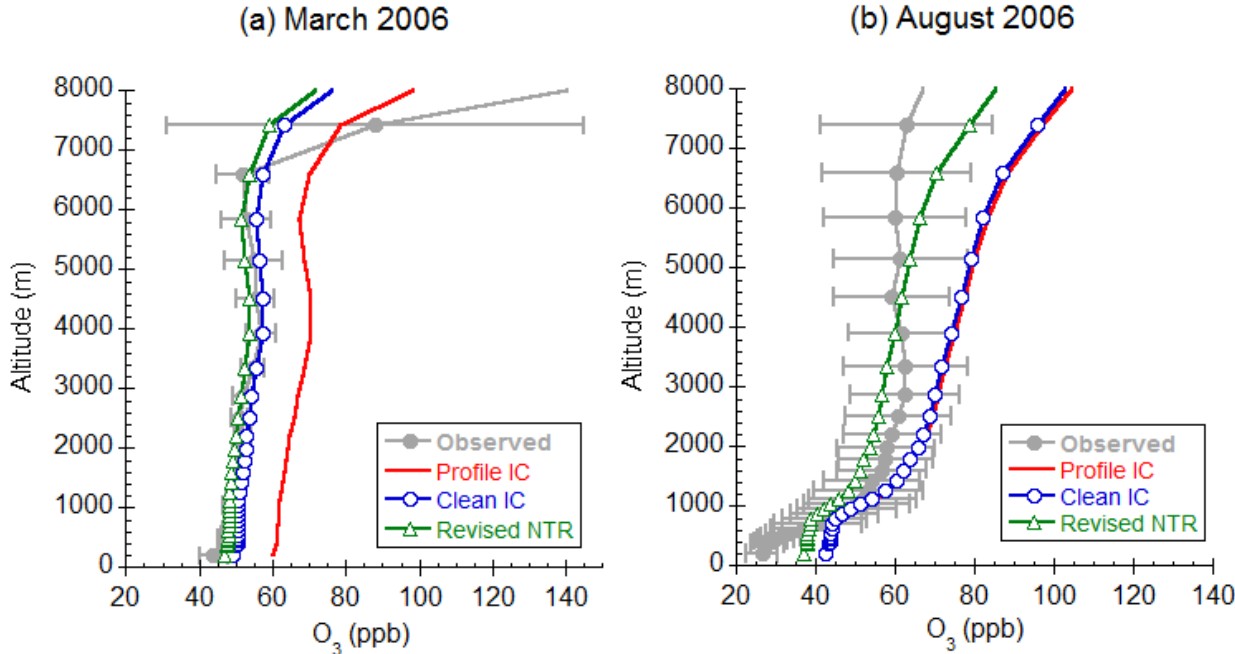

**Figure 4**: Comparisons of simulated average vertical profiles of $O_3$ with ozonesonde measurements at Trinidad Head, California, USA: (a) March 2006, and (b) August 2006. Also shown is ±1 standard deviation of the observed mixing ratios. "Profile IC" uses the default profile for initialization as in regional CMAQ applications, "Clean IC" is the case where the model is spun up from clean conditions, and "Revised NTR" is the simulation with "Clean IC" with updates to the physical and chemical sinks for the species NTR representing organic nitrates.







**Figure 5**: Comparison of simulated 3-D distributions of $O_3$ mixing ratios with observations from the DC8 aircraft during the INTEX-B field campaign: (a-d) comparison of observed $O_3$ and simulated values from various model configurations along flight tracks on specific days; (e) comparison of model and observed campaign average $O_3$ vertical profile for flights over the sub-tropical Pacific during 17 April-1 May 2006; (f) comparison of model and observed campaign average $O_3$ vertical profile for flights over the sub-Arctic Pacific during 1-15 May 2006. Also shown in (e) and (f) is ±1 standard deviation for the observed values.



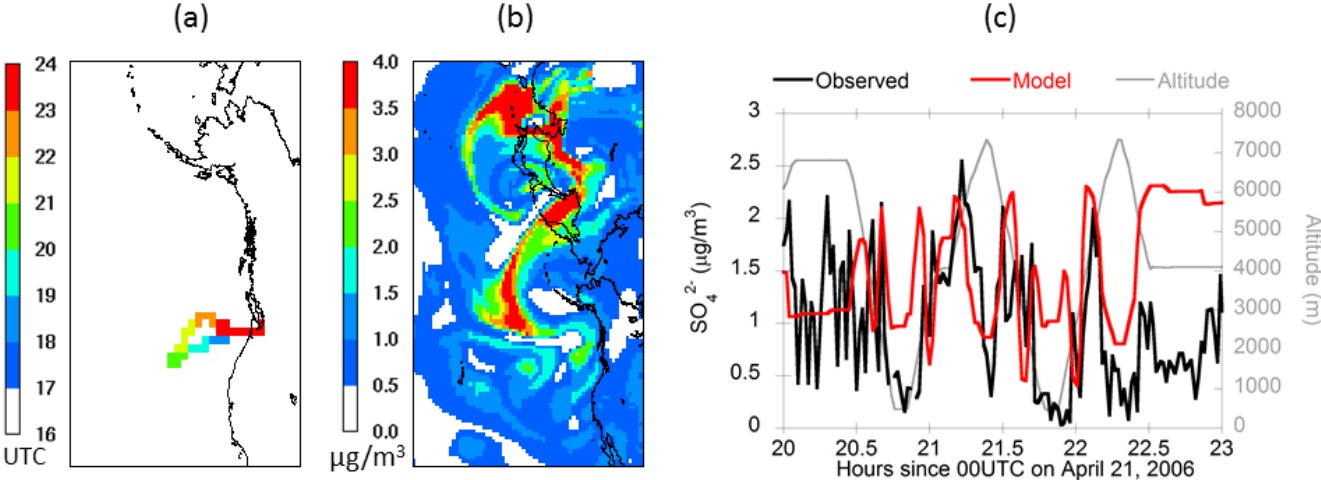

**Figure 6**: Simulation of impact of trans-Pacific transport on 3-D distributions of $SO_4^{2-}$ aerosol on 21 April 2006. (a) Flight path of C-130 aircraft color-coded by hour (UTC). (b) Simulated $SO_4^{2-}$ distribution at 4 km altitude on 21 April 2006 at 2100 UTC. (c) Comparison of modelled and observed $SO_4^{2-}$ aerosol concentrations along C-130 flight path.



**Figure 7**: Comparison of modelled and observed campaign average SO$_4^{2-}$ vertical profiles: (a) against measurements from the DC8 aircraft, and (b) against measurements from the C-130 aircraft. Also shown is ±1 standard deviation for both observed and modelled values.



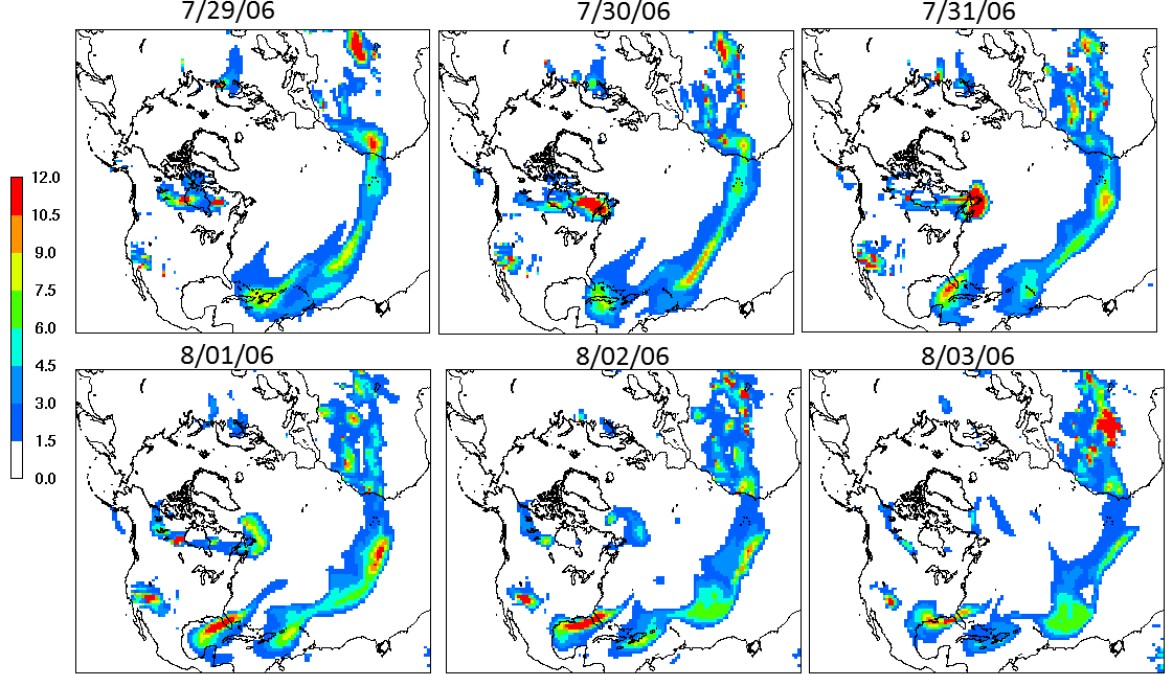

**Figure 8**: Trans-Atlantic transport of a Saharan dust plume, 29 July – 3 August 2006, as simulated by hemispheric CMAQ. Shown in the panels is the difference in daily-average PM$_{2.5}$ concentrations (µg m$^{-3}$) between CMAQ simulations with and without considering dust emissions.



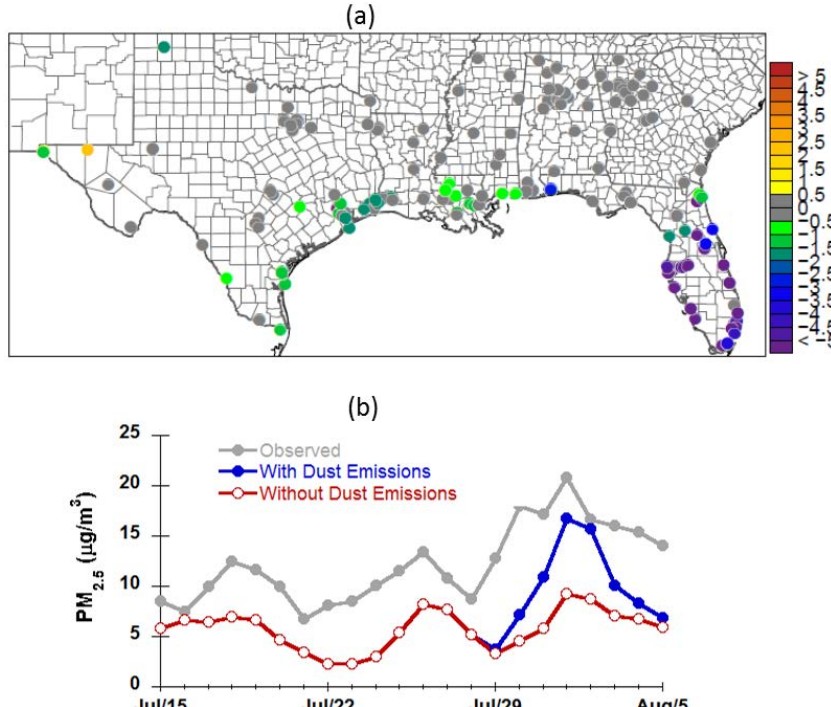

**Figure 9**: The impact of simulated trans-Atlantic transport of Saharan dust on daily-average surface PM$_{2.5}$ in the Gulf region. (a) Average change in bias in daily average surface PM$_{2.5}$ (in μg m$^{-3}$) at AQS monitor locations between CMAQ simulations with and without considering dust emissions, 29 July-2 August 2006. Negative changes in bias denote improvement in model performance by including Saharan dust emissions and representing its trans-Atlantic transport. (b) Comparison of modelled and observed daily-average surface PM$_{2.5}$ averaged over all AQS monitor locations in Florida.





**Figure 10**: Impact of dynamic PV scaling on surface-level seasonal mean maximum daily 8-hour average O$_3$ (MD8O$_3$) mixing ratios (in ppb) estimated as the difference between the simulation with and without the dynamic PV formulation.





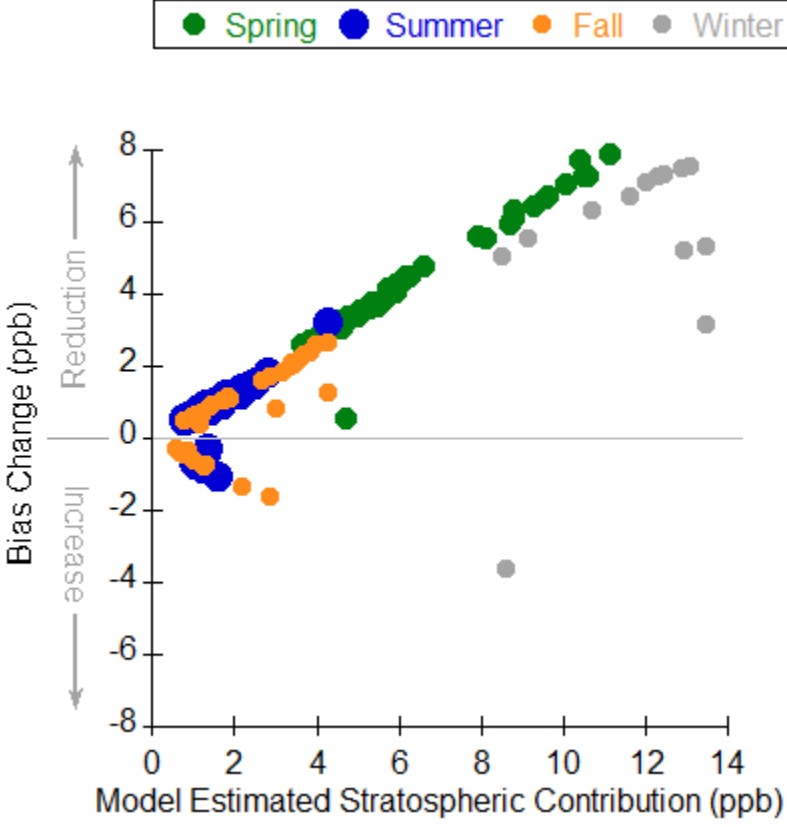

**Figure 11**: Impact of dynamic PV scaling parameterization on model performance for surface-level seasonal mean maximum daily 8-hour average $O_3$ (MD8$O_3$) at CASTNET sites in the U.S. The model estimated stratospheric contribution is estimated as the difference between the simulation with the dynamic PV scaling and one without. The bias change is estimated as the difference between the absolute bias in the simulation with a constant-PV (20 PV) scaling and the absolute bias in the dynamic-PV scaling simulations. Positive changes in bias represent reduction in bias due to dynamic-PV, while negative changes represent an increase in bias in simulated surface MD8$O_3$. Seasonal means are computed based on model-observed pairs when the observed MD8$O_3$ >40 ppb.





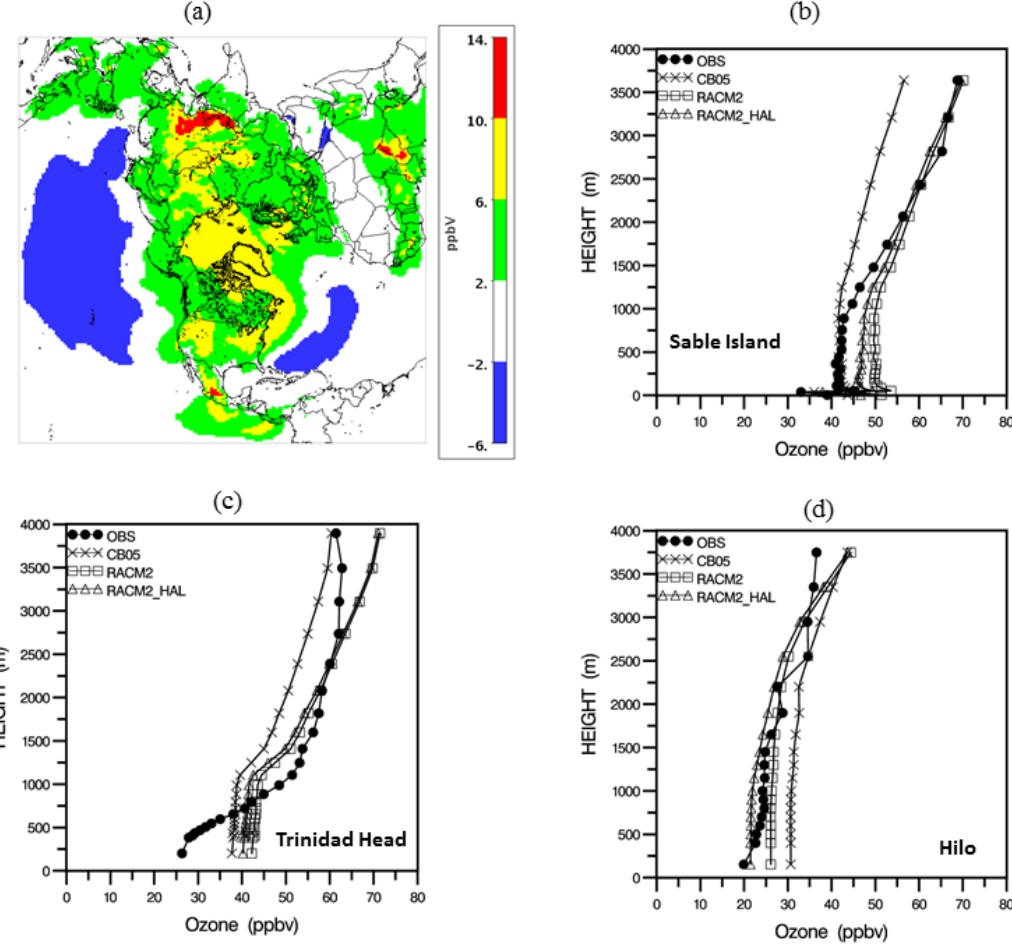

**Figure 12**: Differences in simulated $O_3$ distributions using the RACM2 and CB05TU gas-phase chemical mechanisms in the hemispheric CMAQ model. (a) Mean difference (RACM2 minus CB05TU) in monthly mean surface-level $O_3$ (in ppb) for August 2006. Comparisons of mean (for August 2006) $O_3$ mixing ratio vertical profiles simulated with the CB05TU and RACM2 mechanisms with ozonesonde measurements at (b) Sable Island, Nova Scotia, (c) Trinidad Head, California, and (d) Hilo, Hawaii. RACM2_HAL is from a simulation in which the RACM2 mechanism was augmented with halogen chemistry (described in section 2.4.2).





(a) SCIAMACHY; 2003

(d) CMAQ; 2003

(b) SCIAMACHY; 2010

(e) CMAQ; 2010

(c) SCIAMACHY 2003-2010 Trend

(f) CMAQ 2003-2010 Trend



**Figure 13**: Comparison of observed (left) and model (right) changes in $NO_2$ vertical column density (VCD) across the northern hemisphere. (a) 2003 SCIAMACHY $NO_2$ VCD; (b) 2010 SCIAMACHY $NO_2$ VCD; (c) SCIAMACHY $NO_2$ VCD trend; (d) 2003 CMAQ $NO_2$ VCD; (e) 2010 CMAQ $NO_2$ VCD; (f) CMAQ $NO_2$ VCD trend. VCD is units of $10^{15}$ molecules $cm^{-2}$, and VCD trend is in units of $10^{15}$ molecules $cm^{-2}$ $year^{-1}$.



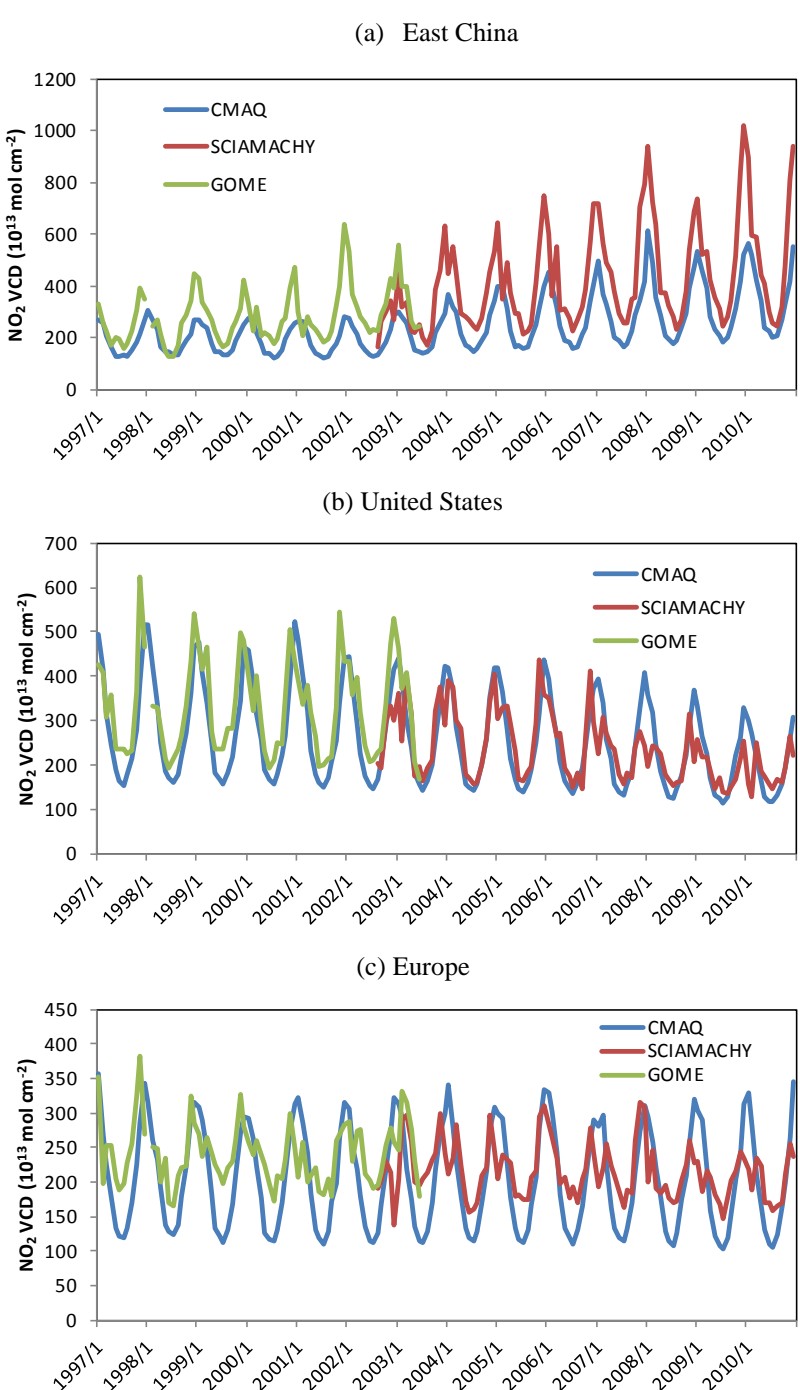

**Figure 14**: Comparison of changes in regional-average modelled NO₂ vertical column density with satellite retrievals from SCIAMACHY and GOME for (a) East Asia, (b) United States, and (c) Europe.





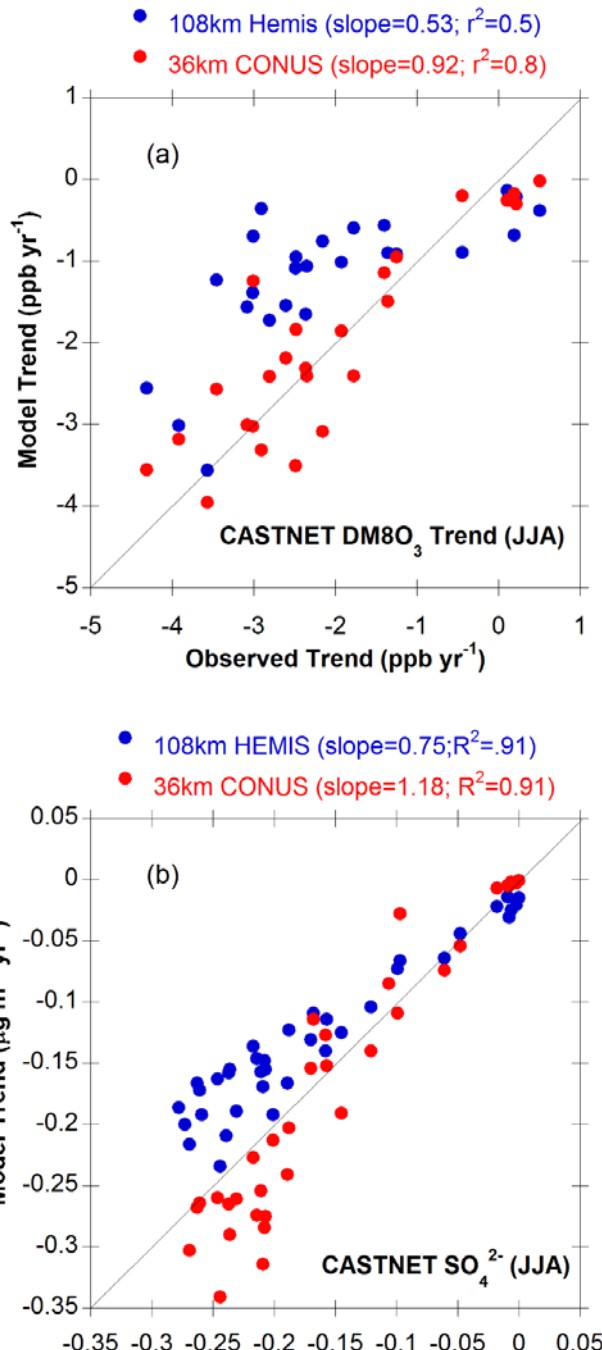

**Figure 15**: Comparison model and observed summertime (JJA) 1990-2010 trends at CASTNET monitoring sites in the U.S. for (a) JJA average daily maximum 8-hour average $O_3$, and (b) JJA average $SO_4^{2-}$. Model results from the 108 km resolution hemispheric CMAQ simulation are shown in blue, while results from a 36-km resolution nested model calculation over the contiguous U.S. are shown in red.





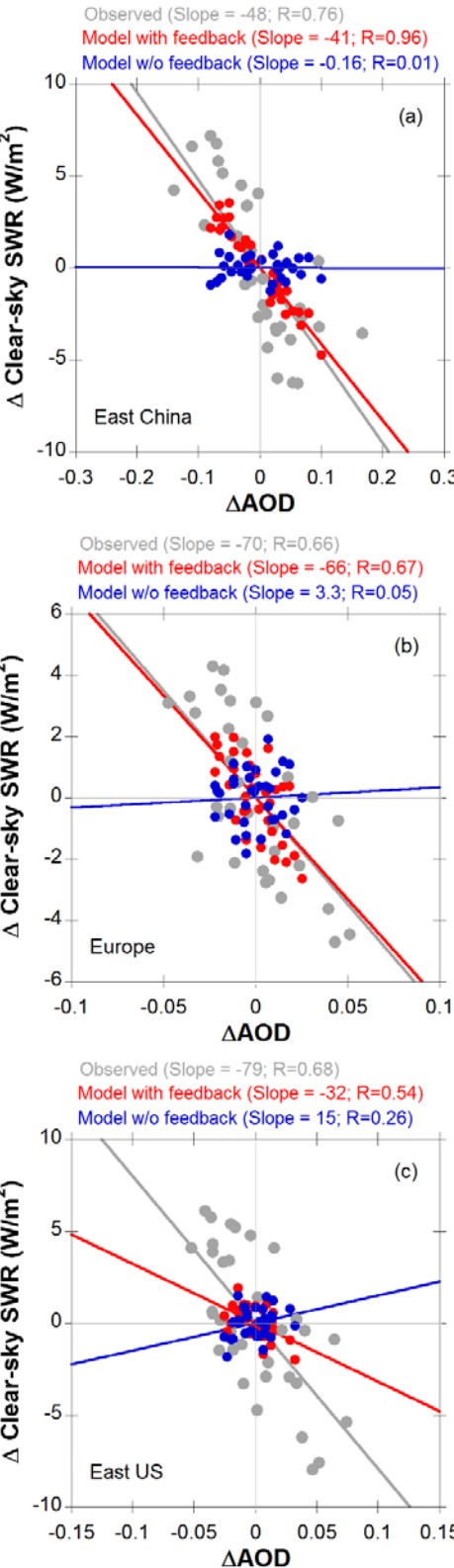





**Figure 16:** Relationship between regional and monthly average (only summer months) changes in aerosol optical depth and changes in surface clear-sky shortwave radiation for (a) East China, (b) Europe, and (c) East U.S. Observed values are shown in grey, CMAQ calculations with direct aerosol radiative feedbacks in red, and CMAQ calculations without aerosol radiative feedbacks in blue. Also, indicated are the slope and correlation coefficient (R) for the individual linear regressions. Changes in AOD and SWR for the 2001-2010 period are relative to the 2000 values.