# Peer review of "Extending the Community Multiscale Air Quality (CMAQ) Modeling System to Hemispheric Scales: Overview of Process Considerations and Initial Applications"

_Atmospheric Chemistry and Physics, 2017_

## Referee Comment (RC1) · Anonymous Referee #2 · 31 May 2017

**General Comments**

This paper provides an overview of the development and some initial applications of a major extension to the off-line Community Multiscale Air Quality (CMAQ) regional chemistry-transport model. Some limitations of regional air quality models are first described and the case is made for the use of hemispheric (or global) air quality models to better address some important research and policy questions. The paper then describes a number of model enhancements that were required to extend CMAQ from its traditional regional-scale configuration to the hemispheric scale, followed by a survey

of a number of different evaluations and applications using this new model version.

This is a well-written paper that describes hemispheric CMAQ, an important enhancement of a widely used regional air quality model to enable it to be applied for larger spatial scales and longer time scales. The process enhancements that were required to achieve this design goal should be of general interest. A diverse set of six different applications of the new hemispheric CMAQ are then presented. Several of these applications have been presented elsewhere while others are presented for the first time. In the former case, however, additional perspective and discussion are provided.

I recommend acceptance of this manuscript with minor revisions. I have made a number of specific comments and suggestions below related to clarity and completeness that I would ask the authors to consider. I have also included a number of editorial comments and corrections that I hope will improve the final version.

**Specific Comments**

1. In Section 2 there were a few places where I asked myself "but what about ...?". In order to provide a more complete description of hemispheric CMAQ, I would suggest that some text could be added to address the following points:

* (Section 2.1) There are many regional chemistry-transport models and there are many global chemistry-transport models, but I am not aware of any other hemispheric chemistry-transport models. Could you add some text to explain the rationale for choosing a hemispheric rather than a global extension, and are there any other models that you are aware of that have also taken your hemispheric approach?

* (Section 2.1) Limited-area models require lateral boundary conditions. Although you have greatly expanded your model domain by choosing a hemispheric domain, hemispheric CMAQ is still a limited-area model. However, there is no discussion in the text of the LBC that you have used for hemispheric CMAQ. Also, Figure 1 showed CMAQ sensitivity to ozone lateral boundary conditions for a regional configuration over the

continental U.S.: has a similar sensitivity test been performed for hemispheric CMAQ to show its sensitivity to LBC specification?

* (Section 2.1) What is the vertical coordinate used by WRF and hemispheric CMAQ?

* (Section 2.2) You recommend here that hemispheric CMAQ should be started from clean tropospheric conditions, and you mention initializing $O_3$ at 30 ppb throughout the model column for the clean IC case. Do you similarly recommend that other CMAQ model species should be set to a uniform clean value in both the horizontal and vertical? Also, can you mention at least some of the clean IC values used for other CMAQ species (e.g., NO, $NO_2$, CO, $NH_3$, $SO_2$, ...)? And given its complex suite of sources, have you examined how CO responds to a 9- or 12-month spin-up from clean conditions?

* (Section 2.3.1) Is the ARCTAS emissions inventory associated with a nominal base year?

* (Section 2.3.1) Are the GEIA biogenic VOC and lightning NOx emissions climatological or year-specifc?

* (Section 2.3.1) Natural emissions take on increased importance for a global or hemispheric chemistry-transport model. Were sea-salt emissions, biomass-burning emissions, soil NO emissions, or volcanic SO2 emissions considered by hemispheric CMAQ?

* (Section 2.3.3) What about marine DMS emissions?

* (Section 2.4.3) It could be mentioned here that RACM2 does not include all (any?) of the chlorine, bromine, and iodine species discussed in Section 2.4.2 – this is relevant to Section 3.4.

* (Section 2.5) In essence this section deals with chemical upper boundary conditions. How are other CMAQ model species treated at the top of the model?

2. It is not made clear in Section 3.1 (p. 15, l. 12) exactly which forms of the two gas-phase chemistry mechanisms were used. Although Sarwar et al. (2013) is referenced (l. 11), I am not sure that the exact mechanism versions used for that paper were also used in this study. For example, I think but I am not sure that a modified version of CB05TU was used in this study that included the modifications described in Sections 2.4.1 and 2.4.2. I am not sure whether or not the version of RACM2 used in this study included the modifications described in Section 2.4.1. The analysis presented in this section does state that two different versions of RACM2 were used, one with the modifications described in Section 2.4.2 and one without those changes, but some additional clarification would be very helpful.

3. In preparing figure panels 13d, 13e, and 13f, was the SCIAMACHY averaging kernel applied to the CMAQ $NO_2$ fields?

4. In Section 3.5, how were the station trends that are shown in Figure 15 calculated?

5. In Section 3.6 how were the CMAQ SWR fields calculated in conjunction with predicted cloud fields?

6. In Section 3.6 I am not sure that the discussion of Figure 16 is completely correct, in particular the following sentence: "The change in the SWR and AOD for each summer month in the 2001-2010 period was estimated relative to the corresponding year-2000 value, and the relationship between these changes is examined in Figure 16 for both model simulations with and without direct aerosol feedback effects." If this were true, then for East China, which has been experiencing dimming over the past decade, wouldn't most of the observed SWR changes be negative, whereas for Europe and the eastern U.S., which have been experiencing brightening over the past decade,

wouldn't most of the observed SWR changes be positive? Instead, the difference patterns in Figure 16 seem to be more consistent with the subtraction of the 11-year monthly means; that is, they are centered.

**Technical Corrections**

The manuscript reads very well but it would still benefit from a careful copyediting to add commas, hyphens, and definite articles in some places but remove them in other places (e.g., change "space and time varying" to "space- and time-varying").

Some acronyms are used but never defined: ECMWF, GFS, ADP, PBL, NOAA, AQS

p. 1, l. 13   WRF is also included as a keyword so it could be defined here on the same page.

p. 2, l. 4   Perhaps "... implementation of the *U.S.* National Ambient Air Quality Standards ..."

p. 2, l. 11   Perhaps "... postulated that in limited-area *chemistry-transport* models, ..."

p. 2, l. 14   Perhaps "... derived from the *global* Integrated Forecasting System of ..."

p. 2, l. 16   hPa is the equivalent SI unit to mb.

p. 2, l. 26   "higher values in the high elevation regions": concentration values or variability values?

p. 2, l. 26   Perhaps "Additionally, higher *contributions from* background levels are estimated"

p. 2, l. 30   "Expectedly" makes me think of "bigly" – perhaps "As expected" would be a better choice.

p. 2, l. 34   Perhaps "... that *other* pollutants with atmospheric lifetimes greater than a

few days ..."

p. 3, l. 17   Perhaps "Section 2 *provides an overview of* the ..."

p. 3, l. 21   Perhaps "*Lastly,* Section 4 summarizes the current model *status* ..."

p. 4, l. 28   Could give a reference for the NCEP/NCAR Reanalysis data set?

p. 4, l. 31   Check weblink (extraneous blank after "V3.1/"?)

p. 5, l. 8   Perhaps "... closer attention to model *chemical* initialization ..."

p. 5, l. 13   Perhaps "... based on the model *emissions*, physics, and chemistry ..."

p. 5, l. 24   Perhaps "... Clean IC and Profile IC cases by August, *nine months after the start of the simulation,* suggests the diminishing impact of initialization ..."

p. 8, l. 33   Should it be Xie et al. (2013)?

p. 9, l. 23   The Goliff et al. (2013) reference is missing.

p. 10, l. 5   "... in the modelled upper tropospher*e*/lower stratospher*e*"

p. 10, l. 29   Perhaps "These *hemispheric $O_3$ fields* can then be used ...".

p. 11, l. 20   There are a number of references to the "Pacific": perhaps some could refer to the "Pacific Ocean" instead.

p. 11, l. 24   `DC-8"

p. 11, l. 25   "... *were* based ... *and* sampled ..."

p. 11, l. 27   There are some unexpected capitalizations, such as Pollution, Spring, Winter, and Continental.

p. 12, l. 33   "Particle Into Liquid Sampler" is the more common usage.

p. 13, l. 23   Perhaps "... air pollutants, *dating* back almost a century"

p. 15, l. 15   Perhaps "... are illustrated in Figure 12a *for August*"

p. 17, l. 1   Perhaps "... in the Savanna region *of Africa* both in ..."

p. 17, l. 2   "SCHIAMACHY" (spelling)

p. 17, l. 21-22   The Figure 15 caption and labels state that the analysis shown is for the summer months only and not all year as stated in this sentence.

p. 17, l. 23   Perhaps "...are results from an additional *21-year* simulation with CMAQ ..."

p. 17, l. 28   The "Conclusions" section states a range for the underestimation (p. 20, l. 12) that should be mentioned here.

p. 18, l. 12-21   As a lead in to the next paragraph, it could be mentioned here that the aerosol optics calculations in WRF-CMAQ include the calculation of AOD.

p. 18, l. 29   Perhaps "... using regional *monthly* averages ..." and "*Eastern* U.S."

p. 18, l. 33   Perhaps "... but AOD at noon (local-time) *for model values* to be consistent ..."

p. 20, l. 11   Perhaps "... at *U.S.* CASTNET monitors, ..."

p. 21, l. 33   IONS and WOUDC are not networks.

p. 29   Figure 1 caption: Perhaps "Impact of *ozone* lateral boundary conditions (LBC) ..."

p. 31   Figure 3: Would it be useful to give associated states or longitudes for these four measurement sites?

p. 32   Figure 4 caption: Would it be useful to indicate the elapsed simulation time for these two panels; that is, four months and nine months after the start of the simulation?

p. 33   Figure 5 caption: Does not mention the aircraft altitude time series in the first

four panels.

p. 43   Figure 14 caption: Perhaps "... changes in *regional- and monthly*-average modeled ..."

---

## Referee Comment (RC2) · Anonymous Referee #1 · 20 Jun 2017

In this article, the authors describe the hemispheric version of the Community Multi-scale Air Quality (CMAQ) modeling system and present a variety of applications of the system for evaluation. In general, the paper is of good quality and should be published with minor revisions as detailed below, although it is a description paper of a new model version and does not contain any truly new science.

Comments

I think the authors should make it clearer in the Introduction that Figure 1 is simply a

characterization of the regional-scale CMAQ model and does not necessarily represent the what actually happens in the real atmosphere. CMAQ is known to be rather diffusive (Emery et al. 2011; Garcia-Menendez et al. 2010; Mathur 2008) and probably does not represent this transport very faithfully. Some readers might be fooled into believing that the fractions presented in Figure 1 are realistic, when they are probably not.

The results of Figure 3 are fascinating. Increasing the vertical resolution from 35 layers to 44 layers substantially reduced ozone profiles in the lower atmosphere. This immediately begs the question as to what would happen if the number of layers was increased to 60 or 70! A pet peeve of this reviewer is that air quality and atmospheric chemistry models are not rigorously evaluated as numerical models. In any basic numerical modeling class, one is taught to increase grid resolutions until the solution converges to a consistent result. This is _never_ done in 3-D atmospheric chemistry modeling! How much different would the results be if this simple numerical procedure was carried out? My guess is quite different.

In the description of the model, the authors in several cases describe what is in the version being presented in this paper, but also describe improvements that are or have been worked on. Examples of this include ... (i) seven NTR species rather than 1; (ii) the marine environment chemistry and deposition; (iii) windblown dust parameterization; and, (iv) the ozone-PV parameterization. In these discussions, it's not always clear what's included in this model version used in this paper and what's just an "advertisement" of the improvements to come in the future.

For Figure 6c, the authors make the dubious statement ... "The comparisons in Figure 6c further show that CMAQ captures the SO4 enhancements in the free troposphere associated with this episodic event." In looking at this figure, I find it very hard to not laugh out loud when reading this sentence! The observed and modeled SO4 values are of the same general magnitude, but don't seem to be correlated at ALL. I think the authors should be more truthful in their comments about this figure.

It is admirable that the authors have implemented the RACM2 chemical mechanism into the hemispheric version of CMAQ, which is likely more suitable than CB05 for larger domain applications. However, the authors should give thought to taking the next step and implementing a mechanism that is even more applicable for domains containing regions remote from major sources. Both CB05 and RACM2 were designed for regional-scale applications where NOx concentrations are relatively large compared to values found in the remote troposphere.

Emery et al. (2011) Atmos. Environ. 45, 7341-7351.

Garcia-Menendez et al. (2010) Atmos. Poll. Res. 1, 239-249.

Mathur (2008) JGR-Atmos.113, D17302, doi:10.1029/2007JD009767.

---

## Short Comment (SC1) · 27 Jun 2017

This is a very interesting paper and an admirable endeavor to extend CMAQ to hemispheric scales. I was especially interested in the handling of alkyl nitrate lifetime. Our group has done similar analysis of alkyl nitrates with comparisons to satellite and aircraft observations in using the CMAQ with CB05 chemistry and CAMx with CB6r2 chemistry (Canty et al., 2015, Goldberg et al., 2016).

What is the lifetime of NTR in this new model framework? Our modification of NTR

so that its lifetime is much shorter (∼1 day), expected if hydroxynitrates are the most abundant species, has led to a better representation of NTR in CB05 when compared to aircraft observations taken during DISCOVER-AQ. Across our model domain, tropospheric column NO2 from CMAQ was in better agreement with satellite observations when the NTR lifetime was decreased. The faster decomposition of NTR also led to an increase in modeled surface ozone. Based on this, the decrease in ozone reported in Fig 4 of this manuscript was a surprise. Is this decrease due to transport or to increased deposition processes? The improved speciation of NTR in the CB6r2 chemical mechanism led to a shorter lifetime of NTR, in the model, without any needed changes to the NTR chemistry.

Great job with this analysis!

- Tim Canty

Canty, T. P., et al, Ozone and NOx chemistry in the eastern US: evaluation of CMAQ/CB05 with satellite (OMI) data, Atmos. Chem. Phys., 15, 10965-10982, doi:10.5194/acp-15-10965-2015, 2015.

Goldberg, D. L., et al., CAMx ozone source attribution in the eastern United States using guidance from observations during DISCOVER-AQ Maryland, Geophys. Res. Lett., 43, 2249–2258, doi:10.1002/2015GL067332, 2016.
* * *

---

## Author Comment (AC1) · 10 Aug 2017

**Response to Comments by Anonymous Referee #2**

*General Comments: This paper provides an overview of the development and some initial applications of a major extension to the off-line Community Multiscale Air Quality (CMAQ) regional chemistry-transport model. Some limitations of regional air quality models are first described and the case is made for the use of hemispheric (or global) air quality models to better address some important research and policy questions. The paper then describes a number of model enhancements that were required to extend CMAQ from its traditional regional-scale configuration to the hemispheric scale, followed by a survey of a number of different evaluations and applications using this new model version.*

*This is a well-written paper that describes hemispheric CMAQ, an important enhancement of a widely used regional air quality model to enable it to be applied for larger spatial scales and longer time scales. The process enhancements that were required to achieve this design goal should be of general interest. A diverse set of six different applications of the new hemispheric CMAQ are then presented. Several of these applications have been presented elsewhere while others are presented for the first time. In the former case, however, additional perspective and discussion are provided.*

*I recommend acceptance of this manuscript with minor revisions. I have made a number of specific comments and suggestions below related to clarity and completeness that I would ask the authors to consider. I have also included a number of editorial comments and corrections that I hope will improve the final version.*

Response: We thank the reviewer for the overall positive assessment of our manuscript. We thank the reviewer for the thorough review and suggestions, the incorporation of which has led to an improved manuscript. Detailed below is our response to the specific reviewer comments and the changes incorporated in the revised manuscript.

*Comment: In Section 2 there were a few places where I asked myself "but what about ...?". In order to provide a more complete description of hemispheric CMAQ, I would suggest that some text could be added to address the following points.*

Response: Detailed below is our response to the specific points raised by the reviewer and the changes incorporated in the revised manuscript.

*Comment: (Section 2.1) There are many regional chemistry-transport models and there are many global chemistry-transport models, but I am not aware of any other hemispheric chemistry-transport models. Could you add some text to explain the rationale for choosing a hemispheric rather than a global extension, and are there any other models that you are aware of that have also taken your hemispheric approach?*

Response: CMAQ's governing equations are cast in a generalized coordinate form which allows the system to accommodate commonly used horizontal map projections (i.e., Lambert conformal, Mercator, and polar stereographic). The polar stereographic projection allows for convenient

representation of a single hemisphere, and thus as already discussed on lines 1-6 of Section 2.1, this formulation and flexibility enables CMAQ to be used on a horizontal domain covering the Northern Hemisphere set on a polar stereographic projection without altering CMAQ or its input/output file structure. This flexibility was the primary motivation to extend CMAQ to hemispheric versus the global scale. However, it should be noted that the hemispheric expansion is only the first step in creating a multi-scale modeling system that spans urban to global scales which is an area of active research and development in our group. To our knowledge, there are a couple of other active efforts on developing and applying hemispheric scale air quality models: (1) the Danish Eulerian Hemispheric Model (DEHM) (Brandt et al., 2012; https://doi.org/10.1016/j.atmosenv.2012.01.011) and (2) a hemispheric version of the CHIMERE model (https://doi.org/10.5194/gmd-10-2397-2017).

To address the reviewer's point, we have modified the discussion in Section 2.1 as follows: "CMAQ's governing three-dimensional equations for species mass conservation and moment dynamics (number, surface area, and volume) describing modes of particulate size distribution are cast in generalized coordinates (cf., Mathur et al., 2005; Byun and Schere, 2006). This formulation allows CMAQ to accommodate horizontal map projections and vertical coordinates from various meteorological models. This flexibility enables CMAQ to be used on a horizontal domain covering the Northern Hemisphere set on a polar stereographic projection (Figure 2a) without altering CMAQ or its input/output file structure. Polar stereographic projections are also used in the Danish Eulerian Hemispheric Model (Brandt et al., 2012) and a hemispheric version of the CHIMERE model (Mailler et al., 2017)".

*Comment: (Section 2.1) Limited-area models require lateral boundary conditions. Although you have greatly expanded your model domain by choosing a hemispheric domain, hemi- spheric CMAQ is still a limited-area model. However, there is no discussion in the text of the LBC that you have used for hemispheric CMAQ. Also, Figure 1 showed CMAQ sensitivity to ozone lateral boundary conditions for a regional configuration over the continental U.S.: has a similar sensitivity test been performed for hemispheric CMAQ to show its sensitivity to LBC specification?*

Response: The reviewer is correct that lateral boundary conditions also need to be specified for the discrete lateral boundaries of the hemispheric domain. To address the reviewer's concern, in the revised manuscript, we have included the following discussion at the end of Section 2.2 (page 6, line 1-9):

"Similar to regional applications, chemical boundary conditions also need to be specified along the discrete lateral boundaries of the hemispheric domain. In current applications, these are set to same values as in the clean IC case discussed above. Note that the boundaries of the hemispheric domain (shown in Figure 2) are in the area encircling the Earth near the equator. Because of the presence of the intertropical convergence zone (ITCZ) in this region, the mixing of air masses originating in the Northern and Southern Hemispheres occurs relatively slowly, with exchange times of typically about 1 year (e.g., Jacob et al., 1987). Since the atmospheric lifetimes of most modelled species are significantly shorter, any impacts of chemical lateral boundary condition

specification are typically confined to the lower latitudes and do not propagate into the domain. Additional model sensitivity tests should however be conducted in the future to quantify any likely seasonal influence of LBC specification on model predictions in lower latitude regions of the Northern Hemisphere".

*Comment: (Section 2.1)  What is the vertical coordinate used by WRF and hemispheric CMAQ?*

Response: Both WRF and CMAQ use the σ-P vertical coordinate. To clarify this point, we have modified the description in section 2.1 to now read "Current WRF and CMAQ hemispheric applications have utilized a horizontal discretization of a 187x187 grid configuration with a grid spacing of 108km and a σ-P vertical coordinate system"

*Comment: (Section 2.2) You recommend here that hemispheric CMAQ should be started from clean tropospheric conditions, and you mention initializing O3 at 30 ppb throughout the  model column for the clean IC case. Do you similarly recommend that other CMAQ model species should be set to a uniform clean value in both the horizontal and vertical? Also, can you mention at least some of the clean IC values used for other CMAQ species (e.g., NO, NO2, CO, NH3, SO2, ...)? And given its complex suite of sources, have you examined how CO responds to a 9- or 12-month spin-up from clean conditions?*

Response: As suggested in the reviewer's comments, the ideal initialization period is species dependent. In our experiments the initial conditions for other chemical species were based on the profiles presented in Byun and Ching (1999), which nominally represent clean tropospheric conditions. We focused our initialization experiments on $O_3$ because of the extensive available measurements on its vertical variations in the troposphere which was used to guide the analysis. To address the reviewer's suggestion on mentioning the initial conditions for other species we point the reader to the profiles in Byun and Ching (1999) by including the following additional sentence on Page 5 (line 20-21): "Initial conditions for all other chemical species were based on clean tropospheric conditions prescribed in Byun and Ching (1999)".

*Comment: (Section 2.3.1)  Is the ARCTAS emissions inventory associated with a nominal base year?*

Response: As stated in the manuscript, the ARCTAS inventory was compiled to support pre-mission planning for the ARCTAS study. As stated on the website for the data, the inventory was compiled using the most recent data available to represent emissions at that time (http://bio.cgrer.uiowa.edu/arctas/emission.html). Thus the data are not associated with a specific base year.

*Comment: (Section 2.3.1) Are the GEIA biogenic VOC and lightning NOx emissions climatological or year-specifc?*

Response: The GEIA biogenic and lightning $NO_x$ emissions represent climatological averages.

The temporalization of the monthly mean biogenic emissions and the annual lightning $NO_x$ emissions to the hourly scale used by CMAQ is described in Xing et al. (2015a) as already mentioned in the manuscript discussion. To address the reviewer's comment, we modified the discussion in Section 2.3.1 as follows: "In applications to date, biogenic VOC (Guenther et al., 1995) and lightning NOx (Price et al, 1997) emissions were obtained from GEIA (Global Emission Inventory Activity; http://www.geiacenter.org). The monthly biogenic VOC emissions were further temporalized to hourly resolution for each simulation day. Monthly lightning $NO_x$ emissions were distributed evenly to each hour of each simulation day. Xing et al. (2015a) further describe the processing of global emission inventories for CMAQ, including temporalization of the annual estimates to hourly model inputs, vertical distributions of anthropogenic and lightning emissions, and speciation of $PM_{2.5}$ and NMVOC emissions to model primary aerosol constituents and gas-phase species"

*Comment: (Section 2.3.1) Natural emissions take on increased importance for a global or hemispheric chemistry-transport model. Were sea-salt emissions, biomass-burning emissions, soil NO emissions, or volcanic SO2 emissions considered by hemispheric CMAQ?*

Response: We agree with the reviewer that representation of the impact of natural emissions on tropospheric composition is important not only for the hemispheric and global scales but also for regions with rapidly declining anthropogenic emissions. Year-specific large-scale biomass burning emissions are included in the EDGAR inventory and have been used in our simulations – this is now explicitly mentioned in Section 2.3.1. We also include parameterizations for natural wind-blown dust emissions (detailed in Section 2.3.2) as well as sea-salt emissions based on Kelly et al. (2010) (already mentioned in Section 2.3.3). The current model simulations however did not include soil NO emissions or volcanic $SO_2$ emissions – this is now clarified in Section 2.3.1 by including the following sentence: "Emissions of NO from soil or $SO_2$ from volcanos are not considered in the applications presented here".

*Comment: (Section 2.3.3) What about marine DMS emissions?*

Response: DMS emissions and chemistry are not yet considered in the hemispheric CMAQ model. We however are currently testing implementations of: (1) a DMS emission parameterization scheme, and (2) modifications to the chemical mechanisms to include DMS oxidation pathways to represent their effects on atmospheric $SO_4^{2-}$ distributions.

*Comment: (Section 2.4.3) It could be mentioned here that RACM2 does not include all (any?) of the chlorine, bromine, and iodine species discussed in Section 2.4.2 – this is relevant to Section 3.4.*

Response: Yes, the base RACM2 mechanism (Goliff et al. 2013) does not include halogen chemistry. We however have included a version of the mechanism augmented with halogen chemistry and discuss those results in section 3.4. We feel that the sentence (Page 16 lines 7-8) already conveys this distinction: "Also shown are predictions with a model configuration in which

the RACM2 mechanism was augmented with the halogen chemistry described in section 2.4.2"

*Comment: (Section 2.5) In essence this section deals with chemical upper boundary conditions. How are other CMAQ model species treated at the top of the model?*

Response: Section 2.5 only discusses the impact of stratosphere-troposphere exchange on $O_3$ distributions. We do not modify other species concentrations in the upper layers with this parameterization. The concentrations of all other species are dictated by the modeled transport and chemistry processes. To be specific to the discussion in this section and to avoid any confusion we have modified the title of section 2.5 to "Representing Impacts of Stratosphere-Troposphere Exchange on $O_3$ Distributions".

*Comment: It is not made clear in Section 3.1 (p. 15, l. 12) exactly which forms of the two gas-phase chemistry mechanisms were used. Although Sarwar et al. (2013) is referenced (l. 11), I am not sure that the exact mechanism versions used for that paper were also used in this study. For example, I think but I am not sure that a modified version of CB05TU was used in this study that included the modifications described in Sections 2.4.1 and 2.4.2. I am not sure whether or not the version of RACM2 used in this study included the modifications described in Section 2.4.1. The analysis presented in this section does state that two different versions of RACM2 were used, one with the modifications described in Section 2.4.2 and one without those changes, but some additional clarification would be very helpful.*

Response: To address the reviewer's comment, in Section 3.1 we have now clarified the mechanisms used by explicitly stating the following: "Note that the simulation "PV, NoHalogen" employed the CB05TU mechanism, while the other two simulations employed a version of the CB05TU mechanism augmented with the halogen chemistry discussed in Section 2.4.2".

We believe the reviewer's query on the RACM2 mechanism is referring to the discussion related to Figure 12 (b-d) in Section 3.4. These figures show results from 3 simulations denoted CB05TU, RACM2, and RACM2_HAL. As discussed in the text and noted in the figure caption RACM2_HAL represents the RACM2 mechanism augmented with halogen chemistry. We see the reviewer's point that it may not be readily apparent whether the simulation denoted CB05TU employed halogen chemistry or not. To address this concern we now explicitly point this out in Section 3.4 as: "Note that the simulations denoted CB05TU and RACM2 did not include representation of halogen chemistry".

*Comment: In preparing figure panels 13d, 13e, and 13f, was the SCIAMACHY averaging kernel applied to the CMAQ NO2 fields?*

Response: We did not use the averaging kernel in comparisons with satellite column data because it was not available for the entire 2000-2010 period. Thus the model $NO_2$ column was estimated without any averaging kernel and was just a straight integration of model $NO_2$ through the model column from surface to ~50mb. To address the reviewer's comment, we clarify this in the

discussion in Section 3.5 as: "Note that the model $NO_2$ column is estimated by integrating the predicted $NO_2$ fields through the model column from the surface to ~50hPa and did not utilize an averaging kernel".

*Comment: In Section 3.5, how were the station trends that are shown in Figure 15 calculated?*

Response: The trends are estimated as a linear regression of the June-July-August (JJA or summer) average values of a concentration metric (for $O_3$ JJA average of the daily maximum 8-hr average value and for $SO_4^{2-}$, the JJA average of weekly average data; as also indicated in the Figure caption) at each site for the 21 years – 1990-2010. We modified the discussion in the last paragraph of section 3.5 as: "Figure 15 presents comparisons of model and observed trends in summer average daily maximum 8-hour average $O_3$ and summer average weekly-average $SO_4^{2-}$ at each CASTNET monitor site. Trends are estimated as the slope of the linear regression of these concentration metrics for the 21-year period".

*Comment: In Section 3.6 how were the CMAQ SWR fields calculated in conjunction with predicted cloud fields?*

Response: Clear and all-sky shortwave radiation fields can be obtained from the RRTMG radiation scheme used in the WRF-CMAQ modeling system. The aerosol direct feedbacks modify these fields in the 2-way coupled WRF-CMAQ configuration and the difference between these is being analyzed.

*Comment: In Section 3.6 I am not sure that the discussion of Figure 16 is completely correct, in particular the following sentence: "The change in the SWR and AOD for each summer month in the 2001-2010 period was estimated relative to the corresponding year-2000 value, and the relationship between these changes is examined in Figure 16 for both model simulations with and without direct aerosol feedback effects." If this were true, then for East China, which has been experiencing dimming over the past decade, wouldn't most of the observed SWR changes be negative, whereas for Europe and the eastern U.S., which have been experiencing brightening over the past decade, wouldn't most of the observed SWR changes be positive? Instead, the difference patterns in Figure 16 seem to be more consistent with the subtraction of the 11-year monthly means; that is, they are centered.*

Response: We are grateful to the reviewer for noting this discrepancy between text and what is shown in the Figure. We re-examined our calculations and verified that for this figure we indeed did not estimate the change relative to 2000, as erroneously noted in the original manuscript discussion. The reviewer is correct that the characteristics of the data shown in Figure 16 suggest that it is an anomaly (i.e., the mean value subtracted out). We have updated Figure 16 to indicate that that the relationship examined is between the AOD anomaly and SWR anomaly and also clarified this in Section 3.6 (on page 19) as: "The relationship between these deseasonalized values (or anomaly) of SWR and AOD for each summer month in the 2001-2010 period is examined in Figure 16 for both model simulations with and without direct aerosol feedback effects. Also shown

in Figure 16 is the corresponding observed relationship between similarly estimated AOD anomaly and SWR anomaly derived from retrievals from the MODIS and the Clouds and the Earth's Radiant Energy System (CERES; Wielicki et al., 1998) instruments, respectively".

The caption of Figure 16 has also been updated to: "Relationship between regional and monthly average (only summer months) changes in aerosol optical depth and changes in surface clear-sky shortwave radiation during the 2001-2010 period for (a) East China, (b) Europe, and (c) East U.S. Observed values are shown in grey, CMAQ calculations with direct aerosol radiative feedbacks in red, and CMAQ calculations without aerosol radiative feedbacks in blue. Also, indicated are the slope and correlation coefficient (R) for the individual linear regressions. For each data set (model or observed) there are 33 values, corresponding to each summer month over the 11-year (2001-2010) period. The anomaly is estimated by subtracting the corresponding 11-year average for that month."

*Technical corrections: The manuscript reads very well but it would still benefit from a careful copyediting to add commas, hyphens, and definite articles in some places but remove them in other places (e.g., change "space and time varying" to "space- and time-varying").*

Response: We thank the reviewer for the thorough review and the editorial suggestions, majority of which have been incorporated as detailed below.

*Some acronyms are used but never defined: ECMWF, GFS, ADP, PBL, NOAA, AQS*
Response: The acronyms have now been defined.

*p. 1, l. 13  WRF is also included as a keyword so it could be defined here on the same page.*
We have defined WRF in the abstract as suggested.

*p. 2, l. 4 Perhaps "… implementation of the U.S. National Ambient Air Quality Standards …"*
Limited area models have and are helping with the design and implementation of air quality standards across the world. We have thus left the sentence unaltered.

*p. 2, l. 11  Perhaps "… postulated that in limited-area chemistry-transport models, …"*
We have modified the sentence following the reviewer's suggestion.

*p. 2, l. 14  Perhaps "… derived from the global Integrated Forecasting System of …"*
We have modified the sentence following the reviewer's suggestion.

*p. 2, l. 16  hPa is the equivalent SI unit to mb.*

We have replaced mb with hPa throughout the manuscript.

*p. 2, l. 26 "higher values in the high elevation regions": concentration values or variability values?*

We have modified the sentence to clarify that they are higher normalized concentration values.

*p. 2, l. 26 Perhaps "Additionally, higher contributions from background levels are estimated"*

Since these boundary tracers represent "model background" values, they may not necessarily imply higher fractional contributions to the net concentrations. We thus have left the sentence as is.

*p. 2, l. 30 "Expectedly" makes me think of "bigly" – perhaps "As expected" would be a better choice.*

We use both "As expected" and "expectedly" interchangeably through the manuscript and thus have left the sentence as is.

*p. 2, l. 34 Perhaps "... that other pollutants with atmospheric lifetimes greater than a few days"*

As written the sentence implies "any pollutant" with atmospheric lifetimes greater than a few days, which we feel is more appropriate. Thus we have left the sentence as is.

*p. 3, l. 17 Perhaps "Section 2 provides an overview of the ..."*

We feel the sentence as written is ok, and have left it as is.

*p. 3, l. 21 Perhaps "Lastly, Section 4 summarizes the current model status ..."*

We have added "Lastly" to the sentence but feel "model state" is more appropriate than "model status".

*p. 4, l. 28 Could give a reference for the NCEP/NCAR Reanalysis data set?*

We now include the following reference for the NCEP/NCAR Reanalysis data set:

Kalnay, E., M. Kanamitsu, R. Kistler, W. Collins, D. Deaven, L. Gandin, M. Iredell, S. Saha, G. White, J. Woollen, Y. Zhu, M. Chelliah, W. Ebisuzaki, W. Higgins, J. Janowiak, K.C. Mo, C. Ropelewski, J. Wang, A. Leetmaa, R. Reynolds, R. Jenne, and D. Joseph, 1996: The NCEP/NCAR 40-year reanalysis project. Bull. Amer. Meteor. Soc., 77, 437-471.

*p. 4, l. 31 Check weblink (extraneous blank after "V3.1/"?)*

The "extraneous blank" was put in just top space the sentence wording within the formatting – it should be correct in the final typeset.

*p. 5, l. 8   Perhaps "… closer attention to model chemical initialization …"*

We have modified the sentence following the reviewer's suggestion

*p. 5, l. 13   Perhaps "… based on the model emissions, physics, and chemistry …"*

We have modified the sentence following the reviewer's suggestion.

*p. 5, l. 24   Perhaps "… Clean IC and Profile IC cases by August, nine months after the start of the simulation, suggests the diminishing impact of initialization …"*

We have modified the sentence following the reviewer's suggestion.

*p. 8, l. 33   Should it be Xie et al. (2013)?*

Thank you for catching the typo – we have corrected the year of the publication.

*p. 9, l. 23   The Goliff et al. (2013) reference is missing.*

Thank you for pointing that out. We have added the Goliff et al. (2013) reference.

*p. 10, l. 5   "… in the modelled upper troposphere/lower stratosphere"*

We have modified the sentence following the reviewer's suggestion.

*p. 10, l. 29   Perhaps "These hemispheric O3 fields can then be used …".*

We have modified the sentence following the reviewer's suggestion.

*p. 11, l. 20    There are a number of references to the "Pacific": perhaps some could refer to the "Pacific Ocean" instead.*

We have modified some of the instances as suggested.

*p. 11, l. 24  `DC-8"*

We have modified "DC8" to "DC-8" throughout the manuscript following the reviewer's suggestion.

*p. 11, l. 25   "… were based … and sampled …"*

We feel the sentence as written is fine and thus have not changed it.

*p. 11, l. 27     There are some unexpected capitalizations, such as Pollution, Spring, Winter, and Continental.*

We have corrected the capitalization of "pollution" in this sentence.

*p. 12, l. 33   "Particle Into Liquid Sampler" is the more common usage.*

We have modified the sentence following the reviewer's suggestion.

*p. 13, l. 23   Perhaps "... air pollutants, dating back almost a century"*

We have modified the sentence following the reviewer's suggestion.

*p. 15, l. 15   Perhaps "... are illustrated in Figure 12a for August"*

We have modified the sentence following the reviewer's suggestion.

*p. 17, l. 1   Perhaps "... in the Savanna region of Africa both in ..."*

We have modified the sentence following the reviewer's suggestion.

*p. 17, l. 2   "SCHIAMACHY" (spelling)*

Thank you for pointing out the typo – we have corrected it.

*p. 17, l. 21-22  The Figure 15 caption and labels state that the analysis shown is for the summer months only and not all year as stated in this sentence.*

Thank you for noting this discrepancy. The figure caption is correct but the text erroneously suggested annual average – we have rewritten the sentence to correct this inconsistency.

*p. 17, l. 23     Perhaps "...are results from an additional 21-year simulation with CMAQ"*

We have modified the sentence following the reviewer's suggestion.

*p. 17, l. 28   The "Conclusions" section states a range for the underestimation (p. 20,*

l. 12) that should be mentioned here.

We now mention the quantitative range of underestimation in the discussion, as suggested by the reviewer.

*p. 18, l. 12-21  As a lead in to the next paragraph, it could be mentioned here that the  aerosol*

*optics calculations in WRF-CMAQ include the calculation of AOD.*

We feel it should be apparent to the ACP readers that since aerosol optical properties are calculated in WRF-CMAQ, the AOD can be easily estimated and available. We have thus left the discussion unaltered.

*p. 18, l. 29   Perhaps "... using regional monthly averages ..." and "Eastern U.S."*

We have modified the sentence following the reviewer's suggestion.

*p. 18, l. 33   Perhaps "... but AOD at noon (local-time) for model values to be consistent"*

We have modified the sentence following the reviewer's suggestion.

*p. 20, l. 11   Perhaps "... at U.S. CASTNET monitors, ..."*

We have modified the sentence following the reviewer's suggestion.

*p. 21, l. 33   IONS and WOUDC are not networks.*

We have deleted "networks" following the reviewer's suggestion.

*p. 29     Figure 1 caption: Perhaps "Impact of ozone lateral boundary conditions (LBC)"*

Since our calculations did not include any chemical decay mimicking that of $O_3$ we do not think we should suggest that the calculation necessarily represents impacts of $O_3$ LBCs on simulated $O_3$ concentrations.

*p. 31     Figure 3: Would it be useful to give associated states or longitudes for these four measurement sites?*

We have modified the caption of Figure 3 to also provide the state names for the four sites.

*p. 32   Figure 4 caption: Would it be useful to indicate the elapsed simulation time for these two panels; that is, four months and nine months after the start of the simulation?*

Following the reviewer's suggestion, we now indicate the elapsed simulation time for the two panels.

*p. 33 Figure 5 caption: Does not mention the aircraft altitude time series in the first four panels.*

The figure caption now also indicates that the aircraft altitude is shown in (a-d).

*p. 43 Figure 14 caption: Perhaps "... changes in regional- and monthly-average modeled ..."*

We have modified the caption following the reviewer's suggestion.

---

## Author Comment (AC2) · 10 Aug 2017

**Response to Comments by Anonymous Referee #1**

*Comment: In this article, the authors describe the hemispheric version of the Community Multiscale Air Quality (CMAQ) modeling system and present a variety of applications of the system for evaluation. In general, the paper is of good quality and should be published with minor revisions as detailed below, although it is a description paper of a new model version and does not contain any truly new science.*

Response: We thank the reviewer for the overall positive assessment of our manuscript and for the constructive suggestions for improvements. Detailed below is our response to the specific reviewer comments and the changes incorporated in the revised manuscript. We believe the incorporation of the reviewer suggestions has helped improve the quality of the revised manuscript.

The extension of CMAQ to hemispheric scales required enhancements to both the model's structural and process attributes, to adequately represent the expanded space and time-scales. We do feel that the incorporation, synthesis, and systematic evaluation of these enhancements conveyed in this paper, does represent new modeling science.

*Comment: I think the authors should make it clearer in the Introduction that Figure 1 is simply a characterization of the regional-scale CMAQ model and does not necessarily represent the what actually happens in the real atmosphere. CMAQ is known to be rather diffusive (Emery et al. 2011; Garcia-Menendez et al. 2010; Mathur 2008) and probably does not represent this transport very faithfully. Some readers might be fooled into believing that the fractions presented in Figure 1 are realistic, when they are probably not.*

Response: Figure 1 illustrates the impact of lateral boundary conditions on the *simulated* free tropospheric and surface concentration variability. The discussion explicitly states that Figure 1 illustrates the influence of "LBC specification on simulated surface-level concentrations across a typical regional modeling domain covering the contiguous U.S.". The subsequent sentences then describe the model tracer species calculations that were used in constructing the Figure. The figure caption also states that these are model simulated concentrations from CMAQ.

The reviewer's somewhat philosophical question on "what happens in the real atmosphere?" speculates on whether this specific model characterization agrees with those from other models and measurements. The results illustrated in Figure 1 agree with the well-established conceptual understanding (built off both measurements and modeling analysis) of long-range transport where in source regions, pollutants within the boundary layer are convectively lofted to the free troposphere where they are intercontinentally transported over long distances by efficient winds and can subsequently be entrained to the surface (subsidence, cloud and boundary layer mixing) in receptor regions thereby regulating "background" concentrations of the receptor region. This conceptual view also implicitly suggests that surface background concentrations are likely influenced by free tropospheric values since that's where transport is more efficient, and that's what the model tracer concentrations suggest. Of course the extrapolation of the illustrated fractions to a specific atmospheric species would also require consideration of specific chemical and physical sinks for that species, and in our discussion of results we have attempted to be deliberate of that distinction. Nevertheless, the relative importance of specification of freetropospheric LBCs (representative of long-range pollutant) transport in any model characterization of "background" pollution cannot be diminished, and that's what the figure is intended to illustrate.

We reread the three papers the reviewer brought to our attention. Mathur et al. (2008) analyze the impact of long-range transport of pollution from the Alaskan fires through complementary analysis of CMAQ simulations, in-situ surface and aloft measurements, and satellite retrievals and show impact on surface concentrations at distant sites in the continental U.S. We are not aware of any suggestions of impacts of excessive diffusive transport characteristics in that analysis. Garcia-Menendez et al. (2010) describe the implementation of an adaptive grid methodology in CMAQ to better resolve horizontal concentration gradients in plumes as they undergo lateral transport, but relative to a fixed resolution coarser grid. Their results are applicable to any model that utilizes a uniform discretization, but do not necessarily point to a systematic issue in the CMAQ modeling system. Emery et al. (2011) analyzed vertical transport in the CAMx model, found excessive vertical transport in mountainous regions due to a combination of using a low-order advection scheme, coarse vertical resolution, and likely artifacts of translation of dynamical information between different vertical grid structures employed by the meteorological model and their CTM. Though they did not present any analysis of CMAQ based calculations, the discussion in their manuscript speculates that models such as CMAQ may exhibit similar characteristics. The extrapolation of the Emery et al (2011) analysis with a different modeling system to the current CMAQ tracer results presented here is not straightforward given the differences in model formulations employed in these tests (advection in Emery vs advection, turbulent and cloud transport, dry deposition and wet scavenging here), set-up (different grid resolutions and coupling with meteorological model both in terms of layer configuration and process representation) and since significant changes have occurred in the CMAQ modeling system over the past 6 years. We thus respectfully disagree with the reviewer's assertion that "CMAQ is known to be rather diffusive", but acknowledge that model inferences of impacts of vertical transport from advection, turbulent mixing, and cloud transport are influenced by the choice of horizontal and vertical grid structures, accuracy of numerical methods employed, and consistency in coupling of these aspects with the driving dynamical model (as also discussed in Section 2.1 of our manuscript).

*Comment: The results of Figure 3 are fascinating. Increasing the vertical resolution from 35 layers to 44 layers substantially reduced ozone profiles in the lower atmosphere. This immediately begs the question as to what would happen if the number of layers was increased to 60 or 70! A pet peeve of this reviewer is that air quality and atmospheric chemistry models are not rigorously evaluated as numerical models. In any basic numerical modeling class, one is taught to increase grid resolutions until the solution converges to a consistent result. This is _never_ done in 3-D atmospheric chemistry modeling! How much different would the results be if this simple numerical procedure was carried out? My guess is quite different.*

Response: The reviewer raises a pertinent issue on the optimal vertical grid structure that should be employed in atmospheric chemistry models. As indicated by Figure 3, the associated discussion in Section 2.1, and our response to the previous comment, more attention needs to be devoted to the vertical grid structure employed by the models. The 44-layer structure depicted in Figure 2 was judiciously designed to provide higher resolution above the boundary layer (nominally at altitudes

> 2km) and near the tropopause to resolve the sharp gradients in $O_3$ mixing ratios. Clearly, employing 60 or 70 layers will improve the resolution further, but greater impacts will be seen if the additional layers are deployed near the tropopause. We whole heartedly agree with the reviewer that rigorous modeling protocols need to be developed to define appropriate layer configurations and vertical and horizontal grid resolutions for specific model applications, and numerical convergence tests that establish the order of accuracy of the models be conducted on a more routine basis. Pragmatically, this is a challenging endeavor as it is not only application-dependent (for instance, an optimal layer configuration designed specifically for resolving stratosphere-troposphere exchange cases may be different from one for resolving boundary layer venting) and also because some parameters (e.g., mixing height, definition of cloud base and top, representation of wind-shear) depend on the discrete layer structure. We echo the reviewer's concerns in the Summary and Concluding Remarks section by re-emphasizing the need for additional model sensitivity simulations to assess the impact of different "model vertical extent and vertical grid resolution" (see page 21).

*Comment: In the description of the model, the authors in several cases describe what is in the version being presented in this paper, but also describe improvements that are or have been worked on. Examples of this include ... (i) seven NTR species rather than 1; (ii) the marine environment chemistry and deposition; (iii) windblown dust parameterization; and, (iv) the ozone-PV parameterization. In these discussions, it's not always clear what's included in this model version used in this paper and what's just an "advertisement" of the improvements to come in the future.*

Response: Since the manuscript describes the extension of the CMAQ modeling system from the traditional regional scale to the expanded hemispheric scale, we have outlined the process representations in CMAQ and how they were enhanced for the extended space and time scales for hemispheric applications. This manuscript is an overview of a modeling system that has evolved over several years, so results are presented from model applications that spanned the development period and contributed to the extension of the CMAQ system. For instance, the initial constant PV-scaling approach helped in the conduct of the multi-decadal simulations. The analysis and availability of the fields from these simulations then enabled the development of a more robust space and time varying PV-scaling parameterization. Similarly, the analysis of NTR species and the initial simple approach to modulate its atmospheric lifetime prompted the investigation of the expanded NTR scheme as well as examination of the RACM2 mechanism.

We agree with the reviewer that it is important to clearly convey what process state is being examined and how the model configurations across the different applications may differ. To address the reviewer's concern we reviewed the descriptions of the model configuration in each section to ensure that the model/process state used in the specific runs is stated clearly.

*Comment: For Figure 6c, the authors make the dubious statement ... "The comparisons in Figure 6c further show that CMAQ captures the SO4 enhancements in the free troposphere associated*

*with this episodic event." In looking at this figure, I find it very hard to not laugh out loud when reading this sentence! The observed and modeled SO4 values are of the same general magnitude, but don't seem to be correlated at ALL. I think the authors should be more truthful in their comments about this figure.*

Response: The discussion was intended to convey the elevated $SO_4^{2-}$ concentrations at 4-6 km altitudes measured by the C-130 aircraft and previously attributed to long-range transport from Asia. The modeled transport patterns in Figure 6b also suggest the influence of long-range transport. We agree that the model does not do a good job at capturing the space-time variability indicated in the measurements; the coarse (108 km) grid resolution is a likely contributor to such discrepancies. To address the reviewer's concern, in the revised manuscript (Page 13; lines 9-12) we have reworded the discussion as follows: "As illustrated in Figure 6c, $SO_4^{2-}$ levels >1 µg m$^{-3}$ were often measured in the free troposphere. Both the observations and model show these enhanced $SO_4^{2-}$ levels at altitudes of 4-6 km, which in conjunction with the large scale simulated $SO_4^{2-}$ distributions in Figure 6b suggest that CMAQ captures the $SO_4^{2-}$ enhancements in the free troposphere associated with this episodic event. Some discrepancies in space-time matched model and observed concentrations are also apparent in Figure 6c, which likely result from the relatively coarse (108 km) horizontal grid resolution employed in the model calculations".

*Comment: It is admirable that the authors have implemented the RACM2 chemical mechanism into the hemispheric version of CMAQ, which is likely more suitable than CB05 for larger domain applications. However, the authors should give thought to taking the next step and implementing a mechanism that is even more applicable for domains containing regions remote from major sources. Both CB05 and RACM2 were designed for regional-scale applications where NOx concentrations are relatively large compared to values found in the remote troposphere.*

Response: We agree with the reviewer that representation of chemistry of the remote troposphere is an important area of future research. Further evaluation of the current mechanisms options in CMAQ for the remote troposphere is an area of current and future research. In the manuscript discussions we acknowledge the need to represent the chemistry of longer-lived species (e.g., acetone) that are important for chemistry of the upper troposphere (page 21; lines 21-25) and also additional analyses of $NO_y$ partitioning and $HO_x$ predictions from the current and any additional chemical mechanism (Page 16; lines 3-4).

---

## Author Comment (AC3) · 10 Aug 2017

**Response to Short Comment by Tim Canty**

*Comment: This is a very interesting paper and an admirable endeavor to extend CMAQ to hemispheric scales. I was especially interested in the handling of alkyl nitrate lifetime. Our group has done similar analysis of alkyl nitrates with comparisons to satellite and aircraft observations in using the CMAQ with CB05 chemistry and CAMx with CB6r2 chemistry (Canty et al., 2015, Goldberg et al., 2016).*

*What is the lifetime of NTR in this new model framework? Our modification of NTR so that its lifetime is much shorter (~1 day), expected if hydroxynitrates are the most abundant species, has led to a better representation of NTR in CB05 when compared to aircraft observations taken during DISCOVER-AQ. Across our model domain, tropospheric column NO2 from CMAQ was in better agreement with satellite observations when the NTR lifetime was decreased. The faster decomposition of NTR also led to an increase in modeled surface ozone. Based on this, the decrease in ozone reported in Fig 4 of this manuscript was a surprise. Is this decrease due to transport or to increased deposition processes? The improved speciation of NTR in the CB6r2 chemical mechanism led to a shorter lifetime of NTR, in the model, without any needed changes to the NTR chemistry. Great job with this analysis!*

Response: Thank you for the positive comments on the manuscript and the analysis. We are aware of the analysis reported in Canty et al. (2015) and read with interest the work reported in Golberg et al. (2016). As discussed in our manuscript and previously (Schwede and Luecken, 2014; Canty et al., 2015, Appel et al., 2017), a thorough description of the NTR family of gases is needed to accurately represent their atmospheric lifetimes and rate of NOx recycling that eventually will influence $O_3$ distributions on local to hemispheric scales. As indicated in the discussion, both physical and chemical sinks of NTR influence its atmospheric lifetimes, but an important distinction is that chemical sinks recycle back $NO_x$ (on varying time scales) while the physical sinks (wet and dry deposition) do not. Clearly, the relative roles of these two NTR removal processes depend on the form of the organic nitrate species. In our initial implementation, we followed Xie et al. (2013) and modified the rate constant for the NTR+OH reaction to that for isoprene nitrates, since on the hemispheric scales organic nitrates formed from isoprene are the largest contributor to the simulated tropospheric NTR burden. More importantly, the dry deposition velocity for NTR was mapped to that for $HNO_3$ and the Henry's law constant for NTR was also mapped to that of $HNO_3$, thereby enhancing wet scavenging of NTR. Thus, for the calculations based on these assumptions, at least at the surface one can expect the lifetime of NTR to be comparable to $HNO_3$ and on the same order or shorter than what was invoked by increasing its photolysis rate by a factor of 10 in your analysis. However, unlike your approach which recycles the NOx back more rapidly locally, an important distinction is that the enhancement of the physical sinks removes the reactive nitrogen from the atmosphere. Thus on the hemispheric scales, less NOx is recycled (due to lower amounts of NTR) and so there is lower ozone shown in Figure 4.

Recently, in CMAQ we have replaced the single alkyl nitrate species (NTR) in CB05TU with seven species to better capture the range of chemical reactivity and Henry's law constants (and thus the physical sinks) – see Appel et al. (2017, GMD). We believe such an approach helps better regulate the organic nitrate budget (also evaluated through comparisons with wet deposition measurements), increases the amount of $NO_x$ recycled locally (resulting in local increases in $O_3$), but reduces the NTR burden on the larger hemispheric scale and better modulates hemispheric background $O_3$. The impacts of the rate of NOx recycling from organic nitrates is also seen in the RACM2 results presented in Figure 12, which show higher $O_3$ in polluted regions, but lower values are seen in the remote areas relative to the CB05TU mechanism; these differences in part arise from higher rates of $NO_x$ recycling from organic nitrates in RACM2 relative to CB05TU. The recently released version of CMAQ (v5.2) includes the CB6 mechanism with updated organic nitrate chemistry; application and evaluation of the CB6 mechanism over hemispheric scales is currently underway.